# SPECTRAL-GUIDED PHYSICAL DYNAMICS DISTILLATION

**Youjin Kim, Dagyeong Na, Jae Yong Lee** [†], **Junseok Kwon** [†]
Chung-Ang University
Seoul 06974, Republic of Korea
{ju9073,dakunglove,jaeyong,jskwon}@cau.ac.kr

## ABSTRACT

The problem of physical dynamics, which involves predicting the 3D trajectories of particles, is a fundamental task with wide-ranging applications across science and engineering. However, accurately forecasting long-horizon trajectories from initial states remains challenging, due to complex particle interactions and entangled multi-scale dynamics involving both low- and high-frequency components. To address this, we propose a novel knowledge-distillation-based framework, **SGDD (Spectral-Guided Dynamics Distillation)**, which integrates a spectral-guided enhancement to adaptively prioritize key frequency components within a unified spatio-temporal representation. Through knowledge distillation, SGDD leverages future trajectories as privileged information during training, guiding a teacher encoder to generate comprehensive dynamics representations while a student encoder approximates them using only the initial state. This enables the student to generate effective dynamics representations at inference, even without privileged information, thereby enabling accurate long-horizon trajectory prediction. Experimental results on molecule, protein, and human motion datasets demonstrate that our method achieves more accurate and stable long-term predictions than previous physical dynamics models, successfully capturing the complex spatio-temporal structures of real-world systems.

## 1 INTRODUCTION

Physical dynamics refers to the problem of predicting and simulating the 3D trajectories of particles across systems at various scales, such as molecules, proteins, and human joints. This problem is fundamental in a wide range of scientific and engineering applications, including drug design (Reddy et al., 2007), protein engineering (Al-Lazikani et al., 2001), and robotics (Spong et al., 2006). In recent years, it has attracted substantial attention, with numerous studies proposing equivariant neural architectures to better capture the underlying symmetries of physical systems (Satorras et al., 2021; Wu et al., 2023a; Du et al., 2022; Xu et al., 2024; Fuchs et al., 2020; Sun et al., 2024).

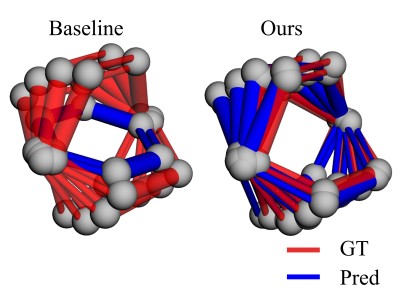

Figure 1: Low-Frequency-Dominated Dynamics. Left: Baseline, Right: Ours.

Despite recent progress, accurately forecasting long-horizon trajectories from initial states remains highly challenging (Lippe et al., 2023). This is because long-horizon prediction amplifies the entanglement of global low-frequency trends and localized high-frequency oscillations. Such entanglement poses significant difficulties for long-term forecasting due to two major contributors: (1) the low- and high-frequency components interplay in complex ways across space and time, and (2) the importance of these components varies across different systems.

---

[†]Corresponding authors.

Considering the first factor, while recent studies often incorporate frequency-aware techniques (Xu et al., 2024) (Sun et al., 2024), they model temporal and spatial structures separately. That is, they rely on spectral representations derived from either temporal or spatial domains in isolation, with limited consideration of their integrated interaction. Consequently, frequency modeling based on a single dimension often struggles to fully capture the underlying physical processes that emerge from the interdependent dynamics of space and time.

For the second contributor, not all frequency components are equally important in long-term prediction. Specifically, low-frequency components capture stable, global patterns, whereas high-frequency components may contribute to instability and noise. Therefore, accurate prediction requires prioritizing low-frequency modes to ensure stability and long-term coherence, while complementarily incorporating high-frequency details to enhance short-term precision.

These challenges are exemplified by Figure 1, where the baseline method struggles to track low-frequency dominant patterns. This underscores the need for frequency-aware spatio-temporal modeling that jointly derives spectral representations from a unified spatio-temporal domain and adaptively emphasizes task-relevant frequency components. To effectively capture the spatio-temporal patterns and key frequency components embedded within entangled dynamics, leveraging trajectory data as privileged information through a knowledge distillation approach can provide direct and efficient guidance.

Building on these, we introduce **SGDD** *(Spectral-Guided Dynamics Distillation)*, a novel knowledge-distillation-based dynamics representation learning framework. *SGDD* leverages future trajectories, which are used only during training as privileged information, to guide the learning of frequency-aware, spatio-temporal dynamics representation. In *SGDD*, a teacher encoder processes trajectories and a student encoder relies only on the initial state, with both producing a spatio-temporal dynamics representation. These representations are then refined through a spectral-guided enhancement module that adaptively emphasizes frequency components most relevant to the target trajectory via learnable weights. Through distillation, the student's enhanced dynamics representation is aligned with the teacher's. This distilled representation serves as an inductive bias for the decoder, enabling accurate long-horizon predictions at inference time, even in the absence of privileged information.

Our main contributions are summarized as follows:

- We propose *SGDD* (**S**pectral-**G**uided **D**ynamics **D**istillation), a novel knowledge distillation framework that extracts rich dynamics representations from privileged future trajectories in the spatio-temporal and spectral domains, and learns to approximate them using only the initial state.

- We introduce a **spectral-guided enhancement module** that refines the dynamics representations by emphasizing key frequency components through learnable weights, thereby providing the decoder with optimized inputs for accurate trajectory prediction.

- We show the effectiveness of *SGDD* on diverse multi-scale particle datasets (MD17, protein, and human motion), where it consistently outperforms strong baselines in trajectory prediction.

## 2 RELATED WORKS

**Physical Dynamics.** Equivariant neural architectures have become essential tools for modeling physical dynamics. EGNN (Satorras et al., 2021) introduced an efficient E(n)-equivariant message passing scheme that jointly updates node features and coordinates. ClofNet (Du et al., 2022) extended this approach by constructing complete local frames to better capture higher-order geometric relations. Attention-based SE(3)-Transformer (Fuchs et al., 2020) ensured SE(3)-equivariance in point clouds and graphs, while Radial Field (Köhler et al., 2019) developed equivariant normalizing flows to enable Boltzmann Generators for symmetry-preserving sampling.

Recent efforts extend beyond spatial equivariance to explicitly address temporal evolution. ESTAG (Wu et al., 2023a) employed an Equivariant DFT together with spatio-temporal modules to capture periodic and non-Markovian behaviors. EGNO (Xu et al., 2024) formulated an Equivariant Graph Neural Operator that directly models trajectories via Fourier-based temporal convolutions. GF-NODE (Sun et al., 2024) integrated Graph Fourier decomposition with Neural ODEs to couple local high-frequency and global low-frequency dynamics.

In contrast to prior studies that emphasize either spatial equivariance or temporal modeling, our work advances them by directly modeling dynamics in the spatio-temporal domain and learning frequency-aware representations that capture long-range structures through privileged supervision.

**Knowledge Distillation using Privileged Knowledge.** Knowledge distillation (KD) was initially introduced for model compression (Hinton et al., 2015), where a teacher network guides a smaller student through soft targets and intermediate features (Tung & Mori, 2019; Shen et al., 2019; Cho & Hariharan, 2019; Yang et al., 2019). Privileged Knowledge (PK), following the Learning Using Privileged Information paradigm (Vapnik & Vashist, 2009), refers to auxiliary signals available only during training, but inaccessible at test time.

Recent studies have extended KD with PK. In human motion prediction, Sun et al. (2022) distilled future poses as PK through a two-step network, enabling the student to exploit privileged supervision while relying solely on observed sequences. In learning-to-rank, Yang et al. (2022) formalized privileged feature distillation, where a teacher model trained with both regular and privileged features transfers knowledge to a student restricted to regular features. Empirical and theoretical analyses demonstrate that KD with PK not only compresses models and improves generalization but also reveals the non-monotonic impact of highly predictive privileged features.

**Graph Knowledge Distillation.** Beyond PK, KD has also been actively studied in graph domains. Although graph neural networks (GNNs) excelled in representation learning, their message passing nature introduced scalability and latency challenges. Graph Knowledge Distillation (Graph KD) addressed this by distilling knowledge from large GNNs to smaller GNNs or lightweight MLPs.

Early GNN-to-GNN distillation methods (Zhang et al., 2020; Yan et al., 2020) reduced parameters but remained constrained by neighborhood-fetching overhead. In contrast, GNN-to-MLP distillation removed explicit message passing, with Zhang et al. (2021) transferring node-level outputs to vanilla MLPs and Wu et al. (2023c) introducing structure-awareness without explicit edges. More recently, Wu et al. (2023b) decomposed teacher knowledge into low- and high-frequency components and injected both into the student MLP, mitigating information-drowning problem and producing distilled MLPs that are both efficient and competitive with GNN teachers.

Together, these two lines of research—KD with privileged knowledge and Graph KD—highlight the versatility of distillation in leveraging auxiliary supervision or structural priors. Building on these insights, we develop a physics-inspired framework that adapts graph-based distillation principles to effectively capture physical dynamics.

## 3 PROPOSED METHOD

### 3.1 TASK DEFINITION AND FRAMEWORK OVERVIEW

**Problem Setting.** We consider the task of multi-step trajectory prediction, which involves forecasting the 3D positions of particles over future time steps. At each time step $t$, the system state is represented as a graph $\mathcal{G}_t = (V, E, \mathbf{Z}_t, \mathbf{h})$, where $V$ denotes the set of $N$ particles (nodes), $E$ represents physical connections between particles, $\mathbf{Z}_t$ is a tensor containing the 3D position $\mathbf{x}_t$ and velocity $\mathbf{v}_t$, and $\mathbf{h}$ encodes node features describing intrinsic physical properties. Given only the initial state $\mathcal{G}_0 = (V, E, \mathbf{Z}_0, \mathbf{h})$, the objective is to predict the sequence of future 3D positions $\{\mathbf{x}_1, \mathbf{x}_2, \dots, \mathbf{x}_T\}$. Our framework performs this prediction by employing a physical dynamics model as the decoder:

$$\{\mathbf{x}_1, \mathbf{x}_2, \dots, \mathbf{x}_T\} = \texttt{Decoder}(\mathcal{G}_0, z), \quad z = \texttt{Encoder}(\mathcal{G}_0). \tag{1}$$

Here, $z$ denotes a node-level dynamics representation, produced by the encoder from the initial state $\mathcal{G}_0$, summarizing the anticipated evolution of physical dynamics. The encoder and decoder are trained jointly in an end-to-end manner, ensuring that the learned representation captures the spatio-temporal patterns essential for accurate trajectory prediction.

**Overview of Proposed Framework.** The core idea of our framework is to construct dynamics representations that assist the decoder in accurately predicting future trajectories. As illustrated in Figure 2, two encoders are employed: the `dynamics encoder` $E_{\text{dyn}}$, which extracts representations $z_{\text{dyn}}$ from the privileged future state sequence $\mathcal{G}_{1:T}$ and initial state $\mathcal{G}_0$, and the `initial encoder` $E_{\text{init}}$, which generates $z_{\text{init}}$ solely from $\mathcal{G}_0$. Both representations are further refined through a spectral-guided enhancement module that leverages a spatio-temporal graph basis to decompose them in

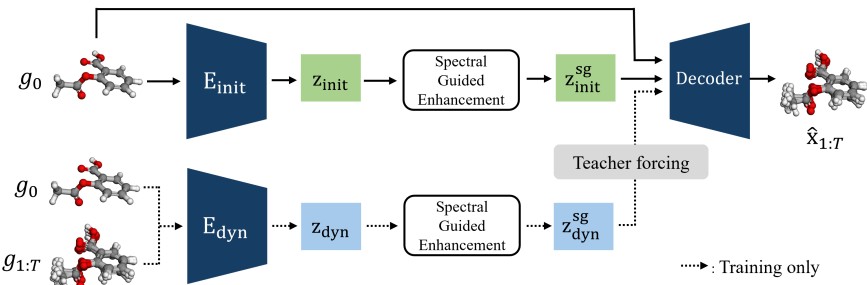

Figure 2: Overall Framework of SGDD.

the spectral domain and adaptively reweight components most relevant to prediction. The resulting spectral-guided representation ($z_{\text{init}}^{sg}$ or $z_{\text{dyn}}^{sg}$) is combined with the initial graph $\mathcal{G}_0$ and passed to the physical dynamics decoder to forecast future trajectories. During training, knowledge distillation is employed so that $z_{\text{init}}^{sg}$ learns to mimic $z_{\text{dyn}}^{sg}$, thereby capturing rich dynamics representations derived from privileged information. At inference, the decoder effectively predicts future trajectories guided by $z_{\text{init}}^{sg}$. The entire framework is trained end-to-end with a staged learning strategy, which ensures stable optimization and effective knowledge distillation.

## 3.2 DYNAMICS REPRESENTATION EXTRACTION

To obtain informative dynamics representations, we construct a spatio-temporal graph from the state sequence, which serves as the structural basis for both the `dynamics encoder` ($E_{\text{dyn}}$), used only during training, and the `initial encoder` ($E_{\text{init}}$), used at inference. Both encoders operate on this graph to produce spatio-temporal representations of particle dynamics.

**Spatio-temporal Graph.** To effectively encode the spatio-temporal information of the state sequence $\{\mathcal{G}_1, \ldots, \mathcal{G}_T\}$, we construct a spatio-temporal graph $\mathcal{G}_{st}$. Each state graph $\mathcal{G}_t = (V, E, Z_t, \text{h})$ shares the same node set and edge set, enabling the formation of a unified graph by linking states along the temporal axis. The resulting $\mathcal{G}_{st}$ combines two independent graphs: a spatial graph $\mathcal{G}_{spatio}$, capturing physical connectivity among particles, and a temporal graph $\mathcal{G}_{temp}$, encoding sequential dependencies across time steps. This spatio-temporal graph $\mathcal{G}_{st}$ serves as the foundation for representing and manipulating dynamics in the spectral domain. For additional details on the construction of the spatio-temporal graph, please refer to Appendix A.2

**Dynamics Encoder** ($E_{\text{dyn}}$). The `dynamics encoder` $E_{\text{dyn}}$ operates on the spatio-temporal graph $\mathcal{G}_{st}$, taking as input the ground-truth state sequence $\{\mathcal{G}_1, \ldots, \mathcal{G}_T\}$ as well as $\mathcal{G}_0$. It processes the spatio-temporal signals to extract a dynamics representation $z_{\text{dyn}}$ that captures both low-frequency components encoding long-term trends and high-frequency components reflecting instantaneous variations:

$$z_{\text{dyn}} = E_{\text{dyn}}(\{\mathcal{G}_1, \ldots, \mathcal{G}_T\}, \mathcal{G}_0), \quad z_{\text{dyn}} \in \mathbb{R}^{N \times T \times d_z}, \tag{2}$$

where $N$ denotes the number of nodes, $T$ the number of time steps, and $d_z$ the dimension of the dynamic representation, respectively.

**Initial Encoder** ($E_{\text{init}}$). The `initial encoder` $E_{\text{init}}$ derives a dynamics representation $z_{\text{init}}$ from the initial state $\mathcal{G}_0$, which contains only spatial edges. To embed $z_{\text{init}}$ in the same spatio-temporal space where $z_{\text{dyn}}$ is defined, we construct an artificial spatio-temporal input by projecting the initial node features through a fully connected layer and expanding them from $\mathbb{R}^{N \times d}$ to $\mathbb{R}^{N \times T \times d}$. This enables the initial encoder to produce representations aligned with the spatio-temporal structure of the dynamics encoder's output. The detailed formulation of the initial encoder is provided in Appendix C.3.2

$$z_{\text{init}} = E_{\text{init}}(\mathcal{G}_0), \quad z_{\text{init}} \in \mathbb{R}^{N \times T \times d_z}. \tag{3}$$

## 3.3 SPATIO-TEMPORAL JOINT BASIS

We define a spatio-temporal joint basis to transform representations into the spectral domain. Let $U_s \in \mathbb{R}^{N \times N}$ and $U_t \in \mathbb{R}^{T \times T}$ be the eigenvector matrices of the normalized Laplacians $L_s$ and $L_t$

for the spatial and temporal graphs, respectively. Specifically, $L_s = U_s \Lambda_s U_s^T$ and $L_t = U_t \Lambda_t U_t^T$, where $\Lambda_s = \text{diag}(\lambda_{s,1}, \lambda_{s,2}, \ldots, \lambda_{s,N})$ and $\Lambda_t = \text{diag}(\lambda_{t,1}, \lambda_{t,2}, \ldots, \lambda_{t,T})$ are diagonal matrices containing the eigenvalues of the respective Laplacians, ordered in ascending order. These eigenvalues represent the frequencies in the spectral domain, with smaller values corresponding to low-frequency (smooth) components and larger values to high-frequency (oscillatory) components. The spatio-temporal joint basis is then constructed via the Kronecker product:

$$B = U_t \otimes U_s, \quad B \in \mathbb{R}^{NT \times NT}. \tag{4}$$

The resulting basis enables projection of spatio-temporal representations into the spectral domain, disentangling complex spatial and temporal frequency components along orthogonal dimensions. Since using a full set of $NT$ basis vectors is computationally expensive, we reduce complexity by selecting the top $K$ modes to form a truncated basis $B_K \in \mathbb{R}^{NT \times K}$. This is achieved by retaining only the columns of $B$ corresponding to the $K$ smallest eigenvalues, defined as:

$$B_K = [b_1, b_2, \ldots, b_K], \quad b_i \in \mathbb{R}^{NT}, \tag{5}$$

where $\{b_i\}$ are the basis vectors associated with the lowest $K$ eigenvalues. This truncation suppresses high-variance, high-frequency content by excluding basis vectors tied to larger eigenvalues. The truncated basis $B_K$ is then employed to project dynamics representations into the spectral domain and reconstruct them back into the spatio-temporal domain, serving as an essential foundation for the spectral-guided enhancement module.

### 3.4 Spectral-Guided Enhancement

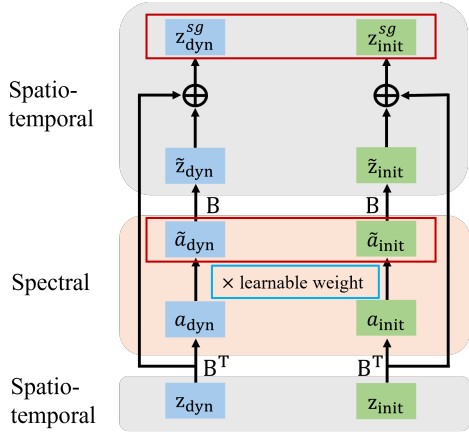

Figure 3: Spectral-Guided Enhancement.

Both the `dynamics encoder` $E_{\text{dyn}}$ and the `initial encoder` $E_{\text{init}}$ produce node-level dynamics representations $z_{\text{dyn}}$ and $z_{\text{init}}$, which are then refined by the spectral-guided enhancement module to adaptively reweight frequency components as shown in Figure 3.

Before describing the enhancement process, we first note how a representation $z \in \mathbb{R}^{N \times T \times d_z}$ (reshaped into $z \in \mathbb{R}^{d_z \times (NT)}$) can be expressed in terms of the spatio-temporal joint basis $B_K \in \mathbb{R}^{NT \times K}$. Since $B_K$ is orthogonal (Appendix B), the matrix $P = B_K B_K^\top$ is the orthogonal projection onto $\text{span}(B_K)$. Thus, any vector $z$ can be decomposed as:

$$z = Pz + (I - P)z, \quad \text{where} \quad P = B_K B_K^\top. \tag{6}$$

Here, $Pz$ denotes the projection within the subspace spanned by $B_K$, while $(I-P)z$ represents the residual component that is orthogonal to it (Appendix B). This decomposition allows $Pz$ to capture the selected spectral modes, while $(I - P)z$ preserves residual information outside the truncated subspace. Having established this decomposition, we now describe the enhancement procedure. We first compute the spectral coefficients by projecting $z$ onto the basis:

$$a := B_K^\top z \in \mathbb{R}^{d_z \times K}. \tag{7}$$

These coefficients are then modulated by learnable, frequency-specific weights $w \in \mathbb{R}^K$ and projected back into the spatio-temporal domain:

$$\tilde{a} := w \odot a, \quad \tilde{z} := B_K \tilde{a} = B_K (w \odot B_K^\top z). \tag{8}$$

Finally, the residual component is added to reconstruct a representation:

$$z^{\text{sg}} := \tilde{z} + (I - P)z. \tag{9}$$

The resulting $z^{\text{sg}}$ integrates the dominant spectral components with residual information, yielding a richer and frequency-aware representation. This formulation provides direct and flexible control in the spectral domain, allowing the model to adaptively emphasize task-relevant frequency bands for improved prediction accuracy.

The outputs of both encoders are processed through this module, producing spectral-guided dynamics representations $z_{\text{dyn}}^{\text{sg}}, z_{\text{init}}^{\text{sg}} \in \mathbb{R}^{N \times T \times d_z}$. In addition, the corresponding spectral coefficients prior to reconstruction are denoted as $\tilde{a}_{\text{dyn}}, \tilde{a}_{\text{init}}$, which are later used for alignment in the spectral domain.

## 3.5 DISTILLATION AND TRAINING STRATEGY

The core objective of our framework is to distill privileged dynamics information $z_{\text{dyn}}$, which is obtained from the future state sequence $\{\mathcal{G}_1, \ldots, \mathcal{G}_T\}$. This information is transferred into $z_{\text{init}}$, which is derived solely from the initial state $\mathcal{G}_0$. This ensures that $z_{\text{init}}$ preserves dynamics-relevant information similar to $z_{\text{dyn}}$, even during inference.

To this end, we enforce dual-level alignment: in the spatio-temporal domain, between $z_{\text{dyn}}^{\text{sg}}$ and $z_{\text{init}}^{\text{sg}}$, and in the spectral domain, between their corresponding coefficients $\tilde{a}_{\text{dyn}}$ and $\tilde{a}_{\text{init}}$. This dual alignment encourages the student representation to capture both low-frequency components that encode stable global trends and high-frequency components that reflect fine-scale variations, thereby improving generalization beyond naive position-level imitation.

The training process adopts a staged learning strategy to ensure stable convergence. In the initial pretraining phase, the teacher forcing ratio is set to 1.0 so that the decoder exclusively receives $z_{\text{dyn}}^{\text{sg}}$ as input. The total loss is defined as

$$\mathcal{L}_{\text{total}} = \mathcal{L}_{\text{pred}}(\mathbf{x}_{1:T}, \hat{\mathbf{x}}_{1:T}) + \lambda \mathcal{L}_{\text{align}}, \tag{10}$$

with the alignment term given by

$$\mathcal{L}_{\text{align}} = \mathcal{L}_{\text{rep}}\left(z_{\text{dyn}}^{\text{sg}}, z_{\text{init}}^{\text{sg}}\right) + \mathcal{L}_{\text{spec}}\left(\tilde{a}_{\text{dyn}}, \tilde{a}_{\text{init}}\right). \tag{11}$$

Here, $\mathcal{L}_{\text{pred}}$ is the mean squared error (MSE) between the predicted and ground-truth trajectories. The alignment loss $\mathcal{L}_{\text{align}}$ consists of representation-level MSE ($\mathcal{L}_{\text{rep}}$) and spectral-level MSE ($\mathcal{L}_{\text{spec}}$). $\mathcal{L}_{\text{align}}$ is weighted by the hyperparameter $\lambda$. To prevent the teacher encoder $E_{\text{dyn}}$ from being influenced or distorted by the student encoder during alignment, gradients are detached from $z_{\text{dyn}}^{\text{sg}}$ and $\tilde{a}_{\text{dyn}}$ when computing $\mathcal{L}_{\text{align}}$.

In the joint training phase, the teacher forcing ratio is set to 0.5, alternating the decoder inputs between $z_{\text{dyn}}^{\text{sg}}$ and $z_{\text{init}}^{\text{sg}}$. In this phase, the `initial encoder` $E_{\text{init}}$ is optimized not only through alignment loss but also via direct supervision from the trajectory prediction loss. The overall loss remains the same as in the pretraining stage.

## 4 EXPERIMENTS

We conducted experiments on molecular dynamics, human motion, and protein datasets to evaluate our SGDD framework. The results show that our method can effectively predict trajectories across diverse systems with different particle scales. In addition, ablation studies confirm that the proposed framework is well-aligned and that individual components make complementary contributions.

**Evaluation Metrics.** Following (Xu et al., 2024), we evaluated performance using two metrics. *State-to-State (S2S)* evaluates only the final state at the last time step. The mean squared error (MSE) loss is computed as $\text{MSE}_{\text{S2S}} = \|\hat{\mathbf{x}}(t_T) - \mathbf{x}(t_T)\|^2$, where $\hat{\mathbf{x}}(t_T)$ denotes the predicted state and $\mathbf{x}(t_T)$ is the ground-truth state at the final timestep $t_T$. *State-to-Trajectory (S2T)* evaluates the entire trajectory by averaging the errors over all $T$ discrete time steps. The loss is defined as $\text{MSE}_{\text{S2T}} = \frac{1}{T} \sum_{k=1}^{T} \|\hat{\mathbf{x}}(t_k) - \mathbf{x}(t_k)\|^2$.

**Baseline.** For the state-to-state (S2S) evaluation, we used the following baselines: SE(3)-Transformer (Fuchs et al., 2020), Tensor Field Networks (Thomas et al., 2018), Message Passing Neural Network (MPNN) (Gilmer et al., 2017), Radial Field (RF) (Köhler et al., 2019), EGNN (Satorras et al., 2021), EGNO (Xu et al., 2024), and GFNODE (Sun et al., 2024). Our proposed SGDD framework was instantiated with different decoder modules, specifically EGNO and GFN-ODE, resulting in two variants: SGDD-EGNO and SGDD-GFNODE. For the state-to-trajectory (S2T) evaluation, we compared against EGNN, EGNO, GFNODE, as well as additional temporal models including NDCN (Zang & Wang, 2020), ITO (Diez et al., 2024), and LG-ODE (Huang et al., 2020).

**Implementation Details** Our framework is implemented in PyTorch, and all experiments are conducted on an NVIDIA A6000 GPU with CUDA 11.6. As the `dynamics encoder`, we employ STSGNN (Chen et al., 2025), which takes the spatio-temporal graph as input, while the `initial`

Table 1: MSE ($\times 10^{-2}$) on MD17 dataset. Upper part: *State-to-State (S2S)*. Lower part: *State-to-Trajectory (S2T)*. The best performance is highlighted in **bold**, the second best is underlined, and performance gains (%) over baselines are shown beneath our SGDD variants.

| S2S | Aspirin | Benzene | Ethanol | Malonaldehyde | Naphthalene | Salicylic | Toluene | Uracil |
|---|---|---|---|---|---|---|---|---|
| RF | 10.94±0.01 | 103.72±1.29 | 4.64±0.01 | 13.93±0.03 | 0.50±0.01 | 1.23±0.01 | 10.93±0.04 | 0.64±0.01 |
| TFN | 12.37±0.18 | 58.48±1.98 | 4.81±0.04 | 13.62±0.08 | 0.49±0.01 | 1.03±0.02 | 10.89±0.04 | 0.84±0.02 |
| SE(3)-Tr. | 11.12±0.06 | 68.11±0.67 | 4.74±0.13 | 13.89±0.02 | 0.52±0.01 | 1.13±0.02 | 10.88±0.06 | 0.79±0.02 |
| EGNN | 14.41±0.15 | 62.40±0.53 | 4.64±0.01 | 13.64±0.01 | 0.47±0.02 | 1.02±0.02 | 11.78±0.07 | 0.64±0.01 |
| EGNN-R | 14.51±0.19 | 62.61±0.75 | 4.94±0.21 | 17.25±0.05 | 0.82±0.02 | 1.35±0.02 | 11.59±0.04 | 1.11±0.02 |
| EGNN-S | 9.50±0.10 | 66.45±0.89 | 4.63±0.01 | 12.88±0.01 | 0.45±0.01 | 1.00±0.01 | 10.78±0.05 | 0.60±0.01 |
| EGNO | 9.18±0.06 | 48.85±0.55 | 4.62±0.01 | 12.80±0.02 | 0.37±0.01 | 0.86±0.02 | 10.21±0.05 | **0.52**±0.02 |
| GFNODE | 7.93±0.00 | 4.82±0.00 | 3.92±0.00 | 12.87±0.00 | 0.37±0.00 | 0.80±0.00 | **4.82**±0.00 | 0.54±0.00 |
| **SGDD-EGNO** | 7.84±0.00 (+14.5%) | 12.97±0.00 (+73.4%) | 4.04±0.00 (+12.5%) | 12.98±0.00 (-1.4%) | 0.36±0.00 (+2.7%) | 0.85±0.00 (+1.1%) | 9.45±0.01 (+7.4%) | 0.53±0.00 (-1.9%) |
| **SGDD-GFNODE** | **7.29**±0.00 (+8.1%) | **2.74**±0.00 (+43.2%) | **3.64**±0.00 (+7.1%) | **12.72**±0.00 (+1.2%) | **0.33**±0.00 (+10.8%) | **0.79**±0.00 (+1.3%) | 5.16±0.00 (-7.1%) | 0.53±0.00 (+1.9%) |

| S2T | | | | | | | | |
|---|---|---|---|---|---|---|---|---|
| NDCN | 31.73±0.40 | 56.21±0.30 | 10.74±0.02 | 46.55±0.28 | 2.25±0.01 | 3.58±0.11 | 13.92±0.02 | 2.38±0.00 |
| ITO | 20.56±0.03 | 57.25±0.58 | 8.60±0.27 | 28.44±0.73 | 1.82±0.17 | 2.48±0.34 | 12.47±0.30 | 1.33±0.12 |
| LG-ODE | 19.36±0.12 | 53.92±1.32 | 7.08±0.01 | 24.41±0.03 | 1.73±0.02 | 3.82±0.04 | 11.18±0.04 | 2.11±0.02 |
| EGNN | 9.24±0.07 | 57.85±2.70 | 4.63±0.00 | 12.81±0.01 | 0.38±0.01 | 0.85±0.00 | 10.41±0.04 | 0.56±0.02 |
| EGNN-R | 12.07±0.11 | 23.73±0.30 | 3.44±0.17 | 13.38±0.03 | 0.63±0.01 | 1.15±0.02 | 5.04±0.02 | 0.89±0.01 |
| EGNN-S | 9.49±0.12 | 29.99±0.65 | 3.29±0.01 | 11.21±0.01 | 0.43±0.01 | 1.36±0.01 | 4.85±0.04 | 0.68±0.01 |
| EGNO | 7.37±0.07 | 22.41±0.31 | 3.28±0.02 | 10.67±0.01 | 0.32±0.01 | 0.77±0.01 | 4.58±0.03 | 0.47±0.01 |
| GFNODE | 6.07±0.09 | 1.51±0.07 | 2.74±0.01 | **9.43**±0.02 | **0.24**±0.02 | 0.63±0.05 | 1.80±0.03 | **0.41**±0.02 |
| **SGDD-EGNO** | 6.20±0.01 (+15.8%) | 7.79±0.18 (+65.2%) | 2.88±0.01 (+12.1%) | 11.01±0.05 (-3.1%) | 0.33±0.00 (-3.1%) | 0.69±0.00 (+10.3%) | 4.23±0.09 (+7.6%) | 0.50±0.00 (-6.3%) |
| **SGDD-GFNODE** | **5.63**±0.01 (+7.2%) | **1.36**±0.01 (+9.9%) | **2.67**±0.01 (+2.5%) | 10.95±0.04 (-16.1%) | 0.26±0.00 (-8.3%) | **0.60**±0.00 (+4.7%) | 2.39±0.03 (-28.8%) | 0.44±0.00 (-4.8%) |

encoder is implemented using GAT (Veličković et al., 2017), which takes the initial state graph as input. The training procedure follows a two-stage strategy: pretraining is performed for approximately one-third of the total epochs, after which joint training is applied for the remaining epochs. Teacher forcing ratios are fixed at 1.0 during pretraining and 0.5 during joint training. The alignment loss weight $\lambda$ is set to 1.0 across all experiments. All models are optimized using the Adam optimizer. Dataset-specific settings (*e.g.*, batch size, learning rate, weight decay, and model configurations) and source code are included in Appendix C.3 and C.4.

## 4.1 MOLECULAR DYNAMICS

**Dataset.** We used MD17 dataset (Chmiela et al., 2017), which provides molecular dynamics trajectories obtained from density functional theory (DFT) simulations. It contains eight small molecules of varying size. Following the same setting as in (Xu et al., 2024), we used 500/2000/2000 random sub-trajectories from the full trajectory of each molecule for training, validation, and testing, respectively. The prediction horizon consists of 8 uniformly spaced timesteps, with the final step corresponding to 3000 frames. The number of atoms corresponding to nodes varies across molecules, typically around 10, and detailed statistics are provided in Appendix C.1.

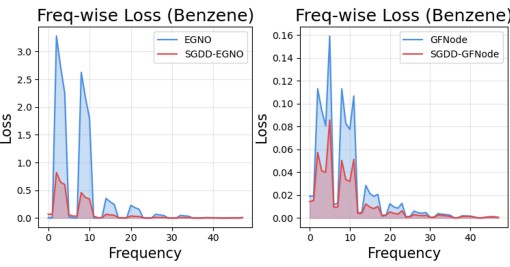

Figure 4: Frequency-wise MSE loss for Benzene.

**Result(Table 1 and Figure 4).** Our SGDD, instantiated as SGDD-EGNO and SGDD-GFNODE, achieved state-of-the-art performance across all molecules in the S2S evaluation, demonstrating its effectiveness in long-horizon trajectory prediction. Notably, for Benzene, SGDD-EGNO shows a 72% performance improvement over EGNO, while SGDD-GFNODE exhibits a 54% improvement over GFNode. This case is significant because both EGNO and GFNODE exhibit high errors concentrated in the low-frequency range. In contrast, our SGDD framework learns dynamics representations that capture low-frequency motion more effectively, thereby providing the decoder with a strong inductive bias and substantially enhancing prediction accuracy as illustrated in Figure 4. In the S2T evaluation, SGDD achieves state-of-the-art results on four molecules (Aspirin, Benzene, Ethanol, Salicylic). For the remaining molecules, the cases where performance falls behind prior models can be attributed to the fact that our framework employs them as decoders, while we cannot fully reproduce their reported results. For a direct comparison with our own implementations of baselines (EGNO, GFNODE), please refer to Appendix D.

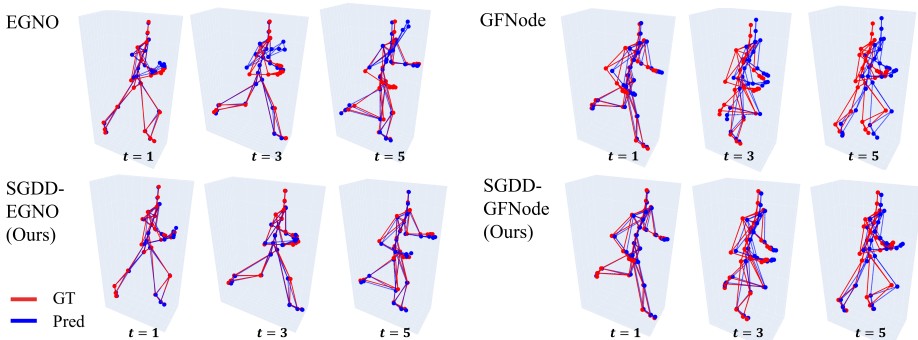

Figure 5: Motion Capture(Run) Visualization.

## 4.2 MOTION CAPTURE

**Dataset.** We utilized the CMU Motion Capture dataset (Carnegie Mellon University, 2003), which consists of 3D skeletal sequences with 31 joints corresponding to nodes. Following the protocol in (Xu et al., 2024), we adopted two representative motion types: walking and running. For the walk and run cases, we constructed 200/600/600 and 200/240/240 sub-trajectories for training, validation, and testing. The prediction horizon consists of 5 uniformly spaced timesteps, with the final step corresponding to 30 frames.

Table 2: MSE ($\times 10^{-2}$) on motion capture dataset. Upper part: *S2S*. Lower part: *S2T*.

| S2S | Walk | Run |
|---|---|---|
| MPNN | 36.1±1.5 | 66.4±2.2 |
| RF | 188.0±1.9 | 521.3±2.3 |
| TFN | 32.0±1.8 | 56.6±1.7 |
| SE(3)-Tr. | 31.5±2.1 | 61.2±2.3 |
| EGNN | 28.7±1.6 | 50.9±0.9 |
| EGNN-R | 90.7±2.4 | 816.7±2.7 |
| EGNN-S | 26.4±1.5 | 54.2±1.9 |
| EGNO | 8.1±1.6 | 33.9±1.7 |
| GFNODE | 9.3±0.0 | 44.9±0.1 |
| **SGDD-EGNO** | 6.7±0.0 (+17.3%) | **28.2**±0.0 (+16.8%) |
| **SGDD-GFNODE** | **6.5**±0.0 (+30.1%) | 31.5±0.1 (+29.8%) |
| S2T | | |
| EGNN-R | 32.0±1.6 | 277.3±1.8 |
| EGNN-S | 14.3±1.2 | 28.5±1.3 |
| EGNO | 3.5±0.5 | 14.9±0.9 |
| GFNODE | 4.5±0.1 | 23.2±2.2 |
| **SGDD-EGNO** | **3.2**±0.0 (+8.6%) | **13.5**±0.9 (+9.4%) |
| **SGDD-GFNODE** | 3.4±0.0 (+24.4%) | 17.7±1.0 (+23.7%) |

**Result (Table 2 Figure 5).** In the S2S evaluation, our framework achieved state-of-the-art performance on both walking and running motions. For these two cases, SGDD-EGNO improved upon EGNO by 17.3% and 16.8%, respectively, while SGDD-GFNODE achieved nearly 30% gains in both cases. In the S2T evaluation, our framework also established new state-of-the-art results across all cases. Specifically, SGDD-EGNO yields 8.6% and 9.4% improvements over EGNO for walking and running, respectively, while SGDD-GFNODE improves upon GFNODE by 24.4% and 23.7%. The larger improvements in the S2T setting suggest that our framework enables more reliable long-horizon representations, leading to stable predictions of human motion dynamics. This is further visually confirmed, where the gap between our method and baselines becomes increasingly evident as the prediction horizon extends.

## 4.3 PROTEIN

Table 3: *S2S* on ADk equilibrium trajectory dataset.

| S2S | |
|---|---|
| Linear | 2.89 |
| RF | 2.84 |
| MPNN | 2.32 |
| EGNN | 2.73 |
| EGHN | 2.03 |
| EGNO | 2.23 |
| EGHNO | 1.80 |
| **SGDD-EGNO** | **1.75** (+21.5%) |

**Dataset.** We used the Adk equilibrium trajectory dataset (Seyler & Beckstein, 2017), which corresponds to the molecular dynamics trajectory of apo adenylate kinase (Gowers et al., 2016). Following the standard setting, we divided the entire trajectory into 2481/827/878 sub-trajectories for training, validation, and testing. The prediction horizon consists of 4 uniformly spaced timesteps, with the final step corresponding to 15 frames. The nodes of the graph are defined as the backbone atoms of the amino acids, resulting in a total of 855 nodes.

**Result (Table 3).** Our SGDD-EGNO achieved a new state-of-the-art with an MSE loss of 1.75, corresponding to a 21.5% improvement over EGNO (2.23). Importantly, the protein dataset contains 855 backbone nodes, forming a relatively large-scale graph. These findings suggest that our framework is capable of learning spectral-guided dynamics representations that provide stable guidance to the decoder for accurate trajectory prediction, even when applied to large-scale spatial systems.

### 4.4 ABLATION STUDIES

Table 4: Ablation study on frequency alignment, feature alignment, and weighting. Results are reported for the SGDD-EGNO model on the MD17 and motion capture (Mocap) datasets. Numbers correspond to the *S2T* metric ($\times 10^{-2}$). The best performance is highlighted in **bold**.

| Freq Align | Feature Align | SGE | Ethanol | Malonaldehyde | Toluene | Mocap-Walk | Mocap-Run |
|:---:|:---:|:---:|:---:|:---:|:---:|:---:|:---:|
| ✓ | ✓ | ✓ | **2.84** | **11.03** | **3.80** | **2.95** | 12.98 |
| ✓ | ✓ | - | 2.90 | 11.04 | 4.18 | 4.04 | **12.61** |
| ✓ | - | ✓ | 2.89 | 11.11 | 4.86 | 3.30 | 13.01 |
| - | ✓ | ✓ | 2.85 | 11.06 | 4.65 | 3.26 | 14.37 |

The central goal of our framework is to construct a dynamics representation that provides the decoder with an effective inductive bias for long-horizon prediction. In a standard encoder–latent representation–decoder pipeline where both inputs and outputs are full future trajectories, the encoder has access to rich dynamical information and can easily produce an informative latent representation. However, in the setting we target—predicting long-horizon trajectories from only the initial state—this privileged supervision is no longer available. A natural way to bridge this gap is to let a teacher encoder observe the future trajectory and let a student encoder learn to approximate the teacher's dynamics representation using only the initial state. This leads to a Knowledge-Distillation-style formulation for learning the dynamics latent representation. Based on this idea, our framework introduces two key design components: (1) Dual Alignment, which aligns teacher–student representations in both the spectral (Freq Align) and spatio-temporal domains (Feature Align), and (2) Spectral-Guided Enhancement (SGE), which adaptively reweights spectral components to emphasize the most informative frequency modes. In the following subsections, we present ablation studies that analyze the contribution and design choices of our framework. Section 4.4.1 evaluates the two core components, Dual Alignment and Spectral-Guided Enhancement. Section 4.4.2 examines the truncation parameter $K$ in the Spectral-Guided Enhancement module. Section 4.4.3 investigates the effect of encoder selection.

#### 4.4.1 ABLATION ON ALIGNMENT AND SPECTRAL GUIDED ENHANCEMENT.

We analyze the contribution of Dual Alignment and the Spectral-Guided Enhancement (SGE) module. As shown in Table 4, removing either component consistently degrades performance across both molecular and human motion datasets. Dual Alignment operates in two complementary domains: (1) Feature Align, which matches teacher–student representations in the spatio-temporal domain to preserve global structural patterns; and (2) Freq Align, which aligns spectral coefficients to capture multi-scale frequency behavior essential for long-horizon stability. There is no consistent superiority between the two alignments alone, but using both simultaneously yields the best performance overall. This suggests that the two alignments operate in a mutually complementary manner, where spatio-temporal alignment provides a robust inductive bias for overall structure, and spectral alignment refines frequency prioritization to mitigate noise and instability. The effect of SGE is observed by comparing the first and second rows: learnable spectral weights through SGE improve performance in nearly all cases by emphasizing informative frequency modes during alignment. Overall, the configuration that includes both alignments and SGE yields the best results, highlighting their complementary roles in forming a robust dynamics representation.

#### 4.4.2 ABLATION ON THE TRUNCATION PARAMETER $K$.

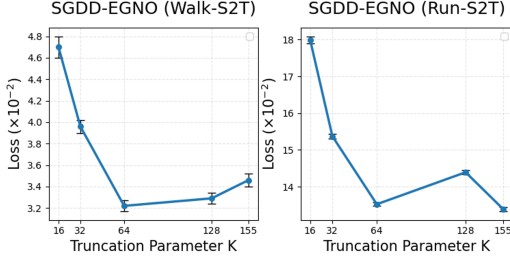

Figure 6: Performance according to truncation parameter $K$.

We further investigated the role of the truncation parameter $K$ in the spatio-temporal joint basis. As described in Section 3.3, we construct a truncated basis $B_K$ by retaining only the $K$ lowest-frequency modes, while the residual $(I - P)z$ preserves information outside this subspace. We conducted experiments on Human Motion dataset with 31 joints and a prediction horizon of 5 timesteps, resulting in a total of $31 \times 5 = 155$ frequency components. As shown in Figure 6, the performance varies with

the choice of the $K$. When $K$ is too small, the enhancement focuses excessively on only lowest-frequency modes. Although the residual term still preserves information outside the truncated subspace, important frequency bands cannot be adequately emphasized. As $K$ increases, the model can adaptively weight a richer set of spectral modes. However, the increase in controllable modes does not always lead to monotonic loss reduction; therefore, it can be needed to select an appropriate K. Experiments on other datasets and discussions on selecting the best $K$ are detailed in Appendix D.2.

### 4.4.3 ABLATION ON ENCODERS

Table 5: MSE ($\times 10^{-2}$) of SGDD-EGNO across different encoder combinations on MD17. (S2S)

| $E_{\text{init}}$ | $E_{\text{dyn}}$ | Aspirin | Ethanol | Naphthalene | Salicylic | Uracil |
|---|---|---|---|---|---|---|
| GAT(142K) | STSGNN(397K) | 7.75 | 4.00 | 0.36 | 0.85 | 0.52 |
| GAT(142K) | STGCN(151K) | 7.97 | 4.08 | 0.36 | 0.84 | 0.52 |
| GINE(336K) | STSGNN(397K) | 7.85 | 4.21 | 0.36 | 0.86 | 0.53 |
| GINE(336K) | STGCN(151K) | 8.02 | 4.15 | 0.36 | 0.86 | 0.52 |
| Transformer(541K) | STSGNN(397K) | 8.90 | 4.12 | 0.36 | 0.91 | 0.53 |
| Transformer(541K) | STGCN(151K) | 8.69 | 4.27 | 0.36 | 0.94 | 0.53 |

We use STSGNN as $E_{\text{dyn}}$ and GAT as $E_{\text{init}}$ in our main experiments, and detailed justifications for these choices are provided in the Appendix C.3. To examine how different encoder choices affect the SGDD framework, we additionally evaluate STGCN (Yan et al., 2018), a spatio-temporal graph convolutional network designed for dynamic skeleton-like structures, as $E_{\text{dyn}}$ and replace GATConv in $E_{\text{init}}$ with either GINEConv(Hu et al., 2019), an edge-enhanced variant of GIN, or Transformer-Conv(Shi et al., 2020), a transformer-based graph convolution layer, on the MD17 dataset. All experiments were conducted under identical training settings. Table 5 reports the performance of different encoder combinations, showing that modifying $E_{\text{init}}$ leads to larger performance variation than modifying $E_{\text{dyn}}$. Replacing GATConv with GINEConv for $E_{\text{init}}$ yields similar results, whereas using TransformerConv leads to lower performance on several molecules. We attribute this to the substantially larger number of learnable parameters in TransformerConv, which leads to overfitting or underfitting under the same training configuration. Overall, SGDD shows a moderate level of robustness to reasonable encoder substitutions—such as GATConv $\leftrightarrow$ GINEConv or STSGNN $\leftrightarrow$ STGCN—while also indicating that heavier encoders may require different optimization settings or regularization strategies to realize their potential. This suggests that SGDD is broadly applicable across encoder architectures, although appropriate training configurations remain important depending on model capacity and dataset scale.

## 5 CONCLUSION

In this work, we introduced SGDD, a novel framework that leverages privileged supervision to learn rich dynamics representations. Our approach combines a spectral-guided enhancement module with a distillation scheme to disentangle multi-scale spatio-temporal dynamics and guide the decoder toward accurate long-horizon trajectory prediction, even at inference time when privileged information is unavailable. Through experiments, we demonstrated that our framework consistently achieves state-of-the-art performance across particle dynamics at different scales. Our study contributes a generalizable framework that unifies spectral representation learning with knowledge distillation, paving the way for future research in robust and scalable physical dynamics modeling. However, our framework also has limitations. Since it relies on existing physical-dynamics models as decoders, its overall performance can be influenced by the capacity of the chosen decoder. Future work could explore decoder-agnostic formulations or tighter integration between representation learning and prediction modules to further enhance robustness and scalability. In addition, SGDD has so far been applied only in settings with fixed spatio-temporal graph structures. Extending it to time-varying graphs—for example, by updating the spatio-temporal basis as the topology evolves or by updating the spatio-temporal basis in an online or adaptive manner—represents a promising future direction for handling dynamic physical environments.

ACKNOWLEDGMENTS

This work was supported in part by the National Research Foundation of Korea(NRF) grant funded by the Korea government(MSIT) (RS-2025-02217071), and in part by the Institute of Information & communications Technology Planning & Evaluation (IITP) grant funded by the Korea government(MSIT) (RS-2021-II211341, RS-2025-25422680).

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

# A NOTATIONS

## A.1 GENERAL NOTATIONS

We summarize the notations used throughout the paper along with their descriptions.

Table A.1: **Summary of Notations**.

| Notation | Description |
|---|---|
| $V$ | Set of $N$ particles (nodes). |
| $E$ | Set of edges representing interactions or connections between particles. |
| $\mathbf{x}_t \in \mathbb{R}^{N \times 3}$ | 3D positions of all $N$ particles at time $t$. |
| $\mathbf{v}_t \in \mathbb{R}^{N \times 3}$ | 3D velocities of all $N$ particles at time $t$. |
| $\mathbf{Z}_t = [x_t, v_t] \in \mathbb{R}^{N \times 6}$ | Node state tensor at time $t$, concatenating position and velocity. |
| $\mathbf{h} \in \mathbb{R}^{N \times d}$ | Time-invariant physical features of each particle. |
| $\mathcal{G}_t = (V, E, \mathbf{Z}_t, \mathbf{h})$ | Graph representation of the system at time step $t$. |
| $\{\mathbf{x}_1, \mathbf{x}_2, \ldots, \mathbf{x}_T\} \in \mathbb{R}^{T \times N \times 3}$ | 3D trajectory of $N$ particles over $T$ time steps. |
| $\{\mathcal{G}_1, \ldots, \mathcal{G}_T\}$ | Future state sequence to be predicted given the initial state $\mathcal{G}_0$. |
| $z_{\text{dyn}}, z_{\text{init}} \in \mathbb{R}^{N \times T \times d_z}$ | Spatio-temporal dynamics representations generated by the dynamics encoder and initial encoder, respectively. |
| $a_{\text{dyn}}, a_{\text{init}} \in \mathbb{R}^{d_z \times K}$ | Spectral coefficients obtained by projecting the dynamics representations onto the joint basis $B \in \mathbb{R}^{NT \times K}$. |
| $w \in \mathbb{R}^K$ | Learnable weights for each spectral mode. |
| $\tilde{a}_{\text{dyn}}, \tilde{a}_{\text{init}} \in \mathbb{R}^{d_z \times K}$ | Frequency-adjusted spectral coefficients after applying the learned weights. |
| $\tilde{z}_{\text{dyn}}, \tilde{z}_{\text{init}} \in \mathbb{R}^{d_z \times NT}$ | Reconstructed representations in the spatio-temporal domain obtained from the adjusted spectral coefficients. |
| $z_{\text{dyn}}^{\text{sg}}, z_{\text{init}}^{\text{sg}} \in \mathbb{R}^{d_z \times NT}$ | Final spectral-guided dynamics representations incorporating residual components: $z_{\text{dyn}}^{\text{sg}} = \tilde{z}_{\text{dyn}} + (I - P)z_{\text{dyn}}$, $z_{\text{init}}^{\text{sg}} = \tilde{z}_{\text{init}} + (I - P)z_{\text{init}}$, where $P = BB^{\top}$. |

## A.2 SPATIO-TEMPORAL GRAPH CONSTRUCTION

In our formulation, the spatio-temporal graph $\mathcal{G}_{st}$ describes both particle interactions and temporal evolution by linking particles within each time step as well as across adjacent time steps. This structure can be viewed as the combination of a spatial graph that captures particle connectivity and a temporal graph that captures sequence order, each admitting its own Laplacian and spectral basis.

**Spatial graph.** The spatial graph $\mathcal{G}_{spatio} = (V, E_s)$ is fixed across time and encodes the physical connectivity among particles. Its normalized Laplacian $L_s$ yields the spatial eigenbasis $U_s$.

**Temporal graph.** The temporal graph $\mathcal{G}_{temp} = (V_t, E_t)$ is defined over the sequence of time indices $V_t = \{1, \ldots, T\}$. To model temporal continuity, we adopt the standard 1D chain construction in which each time step is connected to its previous and next steps:

$$E_t = \{(i, i+1) \mid 1 \leq i < T\}.$$

The normalized Laplacian $L_t$ produces the temporal eigenbasis $U_t$.

# B PROOFS

To apply the Orthogonal Projection Theorem in the spectral domain, we first establish that the truncated spatio-temporal joint basis $B$, constructed from the Laplacian eigenvectors, forms an orthogonal set.

**Proposition B.1** (Orthogonality of the Joint Basis). *Let $U_s \in \mathbb{R}^{N \times N}$ and $U_t \in \mathbb{R}^{T \times T}$ be orthogonal eigenvector matrices of the normalized Laplacians $L_s$ and $L_t$, respectively. Their Kronecker product*

$$B = U_t \otimes U_s \in \mathbb{R}^{NT \times NT}$$

*is also orthogonal, i.e.,*

$$B^{\top} B = I_{NT}.$$

*Consequently, any truncated basis $B_K \in \mathbb{R}^{NT \times K}$ obtained by selecting $K$ columns of $B$ forms a partial orthogonal basis and satisfies $B_K^{\top} B_K = I_K$.*

*Proof.* We use the property of Kronecker products:

$$(U_t \otimes U_s)^\top (U_t \otimes U_s) = (U_t^\top U_t) \otimes (U_s^\top U_s).$$

Since $U_t$ and $U_s$ are orthogonal, $U_t^\top U_t = I_T$ and $U_s^\top U_s = I_N$. Therefore,

$$B^\top B = I_T \otimes I_N = I_{NT}.$$

Selecting a subset of $K$ columns from $B$ preserves orthogonality among the chosen columns, yielding $B_K^\top B_K = I_K$. $\qquad\square$

We next show that the decomposition used in our spectral-guided enhancement is indeed orthogonal.

**Lemma B.2** (Orthogonal Decomposition). *Let $B_K \in \mathbb{R}^{NT \times K}$ be a truncated orthogonal basis and define the projection matrix $P = B_K B_K^\top$. Then, for any $z \in \mathbb{R}^{d_z \times NT}$, the following orthogonal decomposition holds:*

$$z = Pz + (I - P)z, \quad \textit{with} \ \langle Pz, (I - P)z \rangle = 0.$$

*Proof.* Since $P$ is symmetric and idempotent ($P^\top = P$, $P^2 = P$), it is an orthogonal projector. Thus any $z$ can be uniquely decomposed into its projection $Pz$ and residual $(I - P)z$, which are orthogonal because

$$\langle Pz, (I - P)z \rangle = z^\top P^\top (I - P)z = z^\top (P - P^2)z = 0.$$

$\qquad\square$

## C  EXPERIMENT DETAILS

### C.1  DATASET DETAILS

**MD17.** We used the MD17 dataset (Chmiela et al., 2017), which contains molecular dynamics trajectories of eight small molecules. In constructing the graphs, hydrogen atoms are conventionally excluded, and only heavy atoms are retained as nodes, resulting in the number of nodes reported in Table A.2. Edges are defined by extending the original molecular bonds to include up to 2-hop neighbors, following prior work. For each edge, the features are constructed by concatenating the hop type, the atomic types of the connected nodes, and the chemical bond type. Each trajectory is randomly split into train/validation/test sets with 500/2000/2000 state-trajectory pairs, respectively.

Table A.2: Statistics of MD17 dataset.

| Name | Aspirin | Benzene | Ethanol | Malonaldehyde | Naphthalene | Salicylic | Toluene | Uracil |
|---|---|---|---|---|---|---|---|---|
| # nodes | 13 | 6 | 3 | 5 | 10 | 10 | 7 | 8 |

**Motion Capture.** We used the CMU Motion Capture dataset (Carnegie Mellon University, 2003) which contains 3D trajectories of various human motions. Following (Xu et al., 2024), we selected Walk and Run. The data were split into 200/600/600 trajectories for training/validation/testing in Walk, and 200/240/240 trajectories in Run. Each human skeleton is represented as a spatio-temporal graph with 31 joints serving as nodes. Edges are constructed based on the natural skeletal connectivity, i.e., joints directly connected in the human body.

**Protein.** We used the Adk equilibrium trajectory dataset (Seyler & Beckstein, 2017) provided in the MDAnalysis toolkit (Gowers et al., 2016), which contains the molecular dynamics trajectory of apo adenylate kinase. Protein structure is represented as a graph where the nodes correspond to the backbone atoms (N, $C\alpha$, and C of each amino acid), resulting in a total of 855 nodes for apo adenylate kinase. Edges are constructed using a cutoff strategy, where two atoms are connected if their Euclidean distance is within 10 Å in the equilibrium structure, which follows the commonly adopted convention in protein graph construction.

## C.2 BASELINES

We provide an introduction to the two baseline models employed as decoders in our framework.

**EGNO** (Xu et al., 2024) Equivariant Graph Neural Operator (EGNO) is a method for modeling 3D dynamics of relational systems, directly modeling entire trajectory dynamics as temporal functions rather than next-step predictions. It formulates dynamics as a function over time and learns neural operators to approximate it, developing equivariant temporal convolutions in Fourier space stacked over equivariant networks to maintain SE(3)-equivariance while capturing temporal correlations. In terms of frequency-aware approaches in the spectral domain, EGNO performs equivariant temporal convolutions in Fourier space to decompose and model frequency modes, ensuring equivariance in 3D space and handling multi-scale temporal evolution.

**GFNODE** (Sun et al., 2024) Graph Fourier Neural ODEs (GF-NODE) is a neural operator-based model designed to capture spatial-temporal multi-scale interactions in molecular dynamics simulations. It addresses the challenge of predicting long-horizon trajectories by decomposing molecular configurations into spatial frequency modes using the graph Laplacian, evolving these modes in continuous time via Neural ODEs, and reconstructing the updated molecular geometry through an inverse graph Fourier transform. In terms of frequency-aware approaches in the spectral domain, GF-NODE decomposes the spatial structure into frequency components via Graph Fourier Transform, evolves each component temporally via Neural ODEs.

## C.3 MODEL DETAILS

We employed fixed architectures for both the dynamics encoder and the initial encoder throughout all experiments, and in this part, we present a short introduction and implementation details.

### C.3.1 DYNAMICS ENCODER

**Background** We employed STSGNN(Chen et al., 2025) as the backbone for the dynamics encoder. STSGNN takes spatio-temporal graphs as input and performs filtering in the spatio-temporal joint spectral domain. Specifically, based on the eigendecomposition of the normalized Laplacian matrices of spatial and temporal graphs, it introduces the 2-D Discrete Graph Fourier Transform (2-D DGFT) to map input signals into the joint spectral domain. Filtering in this domain is then implemented via bivariate Bernstein polynomial approximation, which leverages learnable coefficients to construct 2-D filters with specialized spectral properties. This formulation enables STSGNN to capture spatial and temporal dependencies simultaneously, unlike conventional methods that rely on separate spatial or temporal spectral representations. Moreover, by exploiting the decoupling property of Bernstein bases, STSGNN can effectively preserve both low- and high-frequency information, while adaptively emphasizing the most task-relevant components. Consequently, it provides stable spatio-temporal joint representations that mitigate instability from high-frequency components while retaining global low-frequency patterns.

**Justification for Choice** STSGNN is a sophisticated spatio-temporal graph encoder that applies 2-D joint spectral filtering, enabling spatial and temporal dependencies to be captured in a single spectral domain while maintaining stable propagation of both low- and high-frequency components. Such frequency-aware modeling aligns well with the nature of physical dynamics, where low-frequency structure governs long-term evolution and high-frequency variations capture short-term fluctuations. Therefore, we adopt STSGNN as our dynamics encoder, as it provides the frequency-resolved spatio-temporal representations required for modeling physical dynamics.

**Implementation Detail**

- **Input:** Spatio-temporal graph node features: $Z_{0:T} \in \mathbb{R}^{N \times T \times 6}$, $h \in \mathbb{R}^{N \times 1 \times d}$
- **Output:** Spatio-temporal dynamics representation $z_{\text{dyn}} \in \mathbb{R}^{N \times T \times d_z}$
- **Detail:** The spatial Bernstein order and temporal Bernstein order were set to the same values as in the original paper, namely 10 and 5, respectively. In addition, the model was configured with 2 layers, an output dimension of 32, a hidden dimension of 32, and a dropout rate of 0.1. The original model was implemented to take multi-step inputs and

generate multi-step outputs. In our framework, we retained only the feature extraction part for multi-step inputs and removed the output module designed for multi-step prediction.

We provide a link to the source code for reproducibility. For specific details, please refer to the following repository[1].

### C.3.2 INITIAL ENCODER

**Background** For the initial encoder, we adopted the Graph Attention Network (GAT) layer (Veličković et al., 2017). GAT extends the idea of message passing in graph neural networks by incorporating a self-attention mechanism over graph neighborhoods. Specifically, instead of treating all neighbors equally or relying on fixed weights, GAT computes attention coefficients that quantify the relative importance of each neighboring node when aggregating features. This is achieved by applying a shared linear transformation to node features, followed by a learnable attention kernel that operates on pairs of nodes. The coefficients are normalized using the softmax function, ensuring that the aggregated representation remains permutation-invariant and adaptive to the underlying graph structure. This formulation allows GAT to capture both local graph topology and feature relevance in a data-driven manner. Compared to spectral approaches that rely on fixed graph filters, GAT provides greater flexibility in learning task-specific dependencies while maintaining computational efficiency.

**Justification for Choice** GAT is the simplest attention-based graph encoder capable of projecting an initial state into a higher-dimensional temporal representation through multi-head attention. In addition, its lightweight architecture and minimal inductive bias help avoid overfitting or underfitting that may arise with heavier encoders, making GAT a suitable choice for initial encoder within our SGDD framework.

**Implementation Detail**

- **Input:** initial state graph node features: $Z_0 \in \mathbb{R}^{N \times 6}$, $\mathrm{h} \in \mathbb{R}^{N \times d}$
- **Output:** Spatio-temporal dynamics representation $z_{\text{init}} \in \mathbb{R}^{N \times T \times d_z}$
- **Detail:** We set the number of attention heads equal to the temporal dimension $T$ to extend the model along the time axis. The hidden dimension was fixed at 32, and we employed 3 layers. The computation proceeds as follows.
  We begin with the initial node features.

$$x_0 = [Z_0, \mathrm{h}] \in \mathbb{R}^{N \times (6+d)}.$$

For a GAT layer with $T$ attention heads, the $t$-th head computes

$$h_{i,t}^{(0)} = W_t^{(0)} x_{0,i},$$

and performs neighborhood aggregation as

$$\tilde{h}_{i,t}^{(0)} = \sum_{j \in \mathcal{N}(i)} \alpha_{ij,t}^{(0)} h_{j,t}^{(0)},$$

where $\alpha_{ij,t}^{(0)}$ denotes the attention coefficient for head $t$.
Concatenating all head outputs yields

$$x_{1,i} = \Big\|_{t=1}^{T} \tilde{h}_{i,t}^{(0)} \in \mathbb{R}^{T d_z}.$$

Thus, the full output of the first layer satisfies

$$x_1 \in \mathbb{R}^{N \times (T d_z)}.$$

For a general $l$-th layer ($l \geq 0$), the update rule is

$$x_{\ell+1,i} = \Big\|_{t=1}^{T} \left( \sum_{j \in \mathcal{N}(i)} \alpha_{ij,t}^{(\ell)} W_t^{(\ell)} x_{\ell,j} \right),$$

---

[1]https://github.com/youjin-DDAI/SGDD

and therefore

$$x_{\ell+1} \in \mathbb{R}^{N \times (Td_z)}.$$

Finally, the output of the last layer,

$$x_L \in \mathbb{R}^{N \times (Td_z)},$$

is reshaped by interpreting the $T$ heads as temporal channels, yielding

$$\mathrm{reshape}(x_L) = z_{\mathrm{init}} \in \mathbb{R}^{N \times T \times d_z}.$$

## C.4 IMPLEMENTATION DETAILS

Our framework is implemented to follow the training and architectural settings of EGNO as closely as possible, and we refer readers to the official EGNO implementation for additional details. For the decoder component, we strictly adopt the architectural configurations of both EGNO and GFNODE without introducing any additional capacity; only training-related hyperparameters are adjusted to accommodate the SGDD training procedure. This ensures that the observed improvements stem from our representation-learning framework rather than increased model complexity in the baseline decoders.

**MD17** We set the batch size to 100, the learning rate to $1 \times 10^{-4}$, and the weight decay to $1 \times 10^{-15}$. Staged learning was adopted, with a maximum of 5000 epochs and pretraining performed up to epoch 2000. The truncation parameter $K$ was fixed at 64 by default. However, when the total number of basis vectors ($N \times T$) for a given molecule was smaller than 64, we set $K = 0.5 \times (N \times T)$ for that molecule, ensuring that $K < N \times T$.

**Motion Capture** We set the batch size to 12, the learning rate to $5 \times 10^{-4}$, and the weight decay to $1 \times 10^{-10}$. The maximum number of epochs was 2000, with pretraining performed up to epoch 500. For the spatio-temporal joint basis, a total of $31 \times 5 = 155$ basis vectors were available. The truncation parameter $K$ was fixed to 64.

**Protein** We set the batch size to 8, the learning rate to $5 \times 10^{-5}$, and the weight decay to $1 \times 10^{-4}$. The maximum number of epochs was 15,000, with pretraining performed up to epoch 100. For the spatio-temporal joint basis, a total of $855 \times 4$ basis vectors were available. In this case, the truncation parameter $K$ was fixed to 128.

# D ADDITIONAL EXPERIMENTAL RESULTS

In this section we provide additional experimental results.

## D.1 COMPARISON OF OUR FRAMEWORK WITH THE IMPLEMENTED BASELINE

We implemented EGNO and GFNODE as baselines and employed them as decoders within our framework. This allows us to compare the performance of the standalone EGNO and GFNODE implementations with their counterparts integrated into our framework, enabling a more direct evaluation. The comparison table on the MD17 dataset is shown in Table A.3 and the comparison table on the motion capture dataset is shown in Table A.4. For GFNODE, since the original model was designed only for the MD17 dataset, the results on the motion capture dataset are identical to those reported in Section 4.2.

## D.2 ABLATION STUDIES

**Ablation on Alignment and Spectral Guided Enhancement.** We conducted ablation studies on both the MD17 datasets using the SGDD-EGNO to examine the effects of different alignment strategies and the presence or absence of learnable weights. The metrics were computed using both S2S and S2T. The results are summarized in Tables A.5 and A.6.

And for SGDD-GFNODE, we conducted only the performance comparison with and without learnable weights, and the experimental results on both the MD17 and motion capture datasets are reported in two tables, corresponding to the S2S and S2T metrics Tables A.7 and A.8.

Table A.3: Comparison on the MD17 dataset with our own implementations. Upper part: *S2S*. Lower part: *S2T*. Best results are highlighted in **bold** and relative improvements (%) of SGDD variants over the corresponding baselines are shown below.

| S2S | Aspirin | Benzene | Ethanol | Malonaldehyde | Naphthalene | Salicylic | Toluene | Uracil |
|---|---|---|---|---|---|---|---|---|
| EGNO | 9.35 | 58.09 | 4.60 | **12.82** | 0.39 | 0.87 | 10.95 | 0.58 |
| **SGDD-EGNO** | **7.75** | **13.65** | **4.00** | 13.08 | **0.36** | **0.85** | **8.34** | **0.52** |
| | (+17.1%) | (+76.5%) | (+13.0%) | (-2.0%) | (+7.6%) | (+2.2%) | (+23.8%) | (+10.3%) |
| GFNODE | 7.64 | 4.89 | 3.92 | 12.86 | 0.37 | 0.80 | 5.00 | 0.54 |
| **SGDD-GFNODE** | **7.38** | **2.23** | **3.64** | **12.72** | **0.34** | **0.79** | **4.97** | **0.52** |
| | (+3.4%) | (+54.3%) | (+7.1%) | (+1.0%) | (+8.1%) | (+1.2%) | (+0.6%) | (+3.7%) |
| **S2T** | | | | | | | | |
| EGNO | 7.03 | 30.79 | 3.27 | **10.83** | 0.35 | 0.75 | 4.86 | 0.54 |
| **SGDD-EGNO** | **6.20** | **8.29** | **2.84** | 11.03 | **0.33** | **0.69** | **3.80** | **0.50** |
| | (+11.8%) | (+73.0%) | (+13.1%) | (-1.8%) | (+5.7%) | (+8.0%) | (+21.8%) | (+7.4%) |
| GFNODE | 6.18 | 2.26 | 2.86 | 11.03 | 0.32 | 0.66 | 2.33 | **0.43** |
| **SGDD-GFNODE** | **5.66** | **1.15** | **2.66** | **10.91** | **0.27** | **0.60** | **2.32** | **0.43** |
| | (+8.4%) | (+49.1%) | (+6.9%) | (+1.0%) | (+15.6%) | (+9.0%) | (+0.4%) | (0.0%) |

Table A.4: Comparison on the motion capture dataset with our own implementations. Upper part: *S2S*. Lower part: *S2T*.

| S2S | Walk | Run |
|---|---|---|
| EGNO | 11.9 | 37.9 |
| **SGDD-EGNO** | **6.7** | **28.2** |
| | (+43.3%) | (+25.5%) |
| **S2T** | | |
| EGNO | 5.5 | 17.6 |
| **SGDD-EGNO** | **3.2** | **13.5** |
| | (+41.5%) | (+23.2%) |

Table A.5: Ablation study on frequency alignment, feature alignment, and weighting. Results are reported for the SGDD-EGNO model on the MD17 datasets. Numbers correspond to the *S2T* metric $(\times 10^{-2})$.

| Freq Align | Feature Align | Weight | Aspirin | Benzene | Ethanol | Malonaldehyde | Naphthalene | Salicylic | Toluene | Uracil |
|---|---|---|---|---|---|---|---|---|---|---|
| ✓ | ✓ | ✓ | 6.20 | 8.29 | 2.84 | 11.03 | 0.33 | 0.69 | 3.80 | 0.50 |
| ✓ | ✓ | - | 6.49 | 6.27 | 2.90 | 11.04 | 0.34 | 0.70 | 4.18 | 0.50 |
| ✓ | - | ✓ | 6.15 | 10.62 | 2.89 | 11.11 | 0.33 | 0.68 | 4.86 | 0.50 |
| - | ✓ | ✓ | 6.26 | 10.79 | 2.85 | 11.06 | 0.33 | 0.69 | 4.65 | 0.50 |

Table A.6: Ablation study on frequency alignment, feature alignment, and weighting. Results are reported for the SGDD-EGNO model on the MD17 datasets. Numbers correspond to the *S2S* metric $(\times 10^{-2})$.

| Freq Align | Feature Align | Weight | Aspirin | Benzene | Ethanol | Malonaldehyde | Naphthalene | Salicylic | Toluene | Uracil |
|---|---|---|---|---|---|---|---|---|---|---|
| ✓ | ✓ | ✓ | 7.75 | 13.65 | 4.00 | 13.08 | 0.36 | 0.85 | 8.34 | 0.52 |
| ✓ | ✓ | - | 8.17 | 11.23 | 4.09 | 12.92 | 0.38 | 0.85 | 9.31 | 0.53 |
| ✓ | - | ✓ | 7.77 | 15.82 | 4.13 | 13.05 | 0.35 | 0.85 | 10.85 | 0.52 |
| - | ✓ | ✓ | 7.90 | 16.55 | 4.07 | 12.96 | 0.36 | 0.86 | 10.36 | 0.52 |

Table A.7: Ablation study on weighting. Results are reported for the SGDD-GFNODE model on the MD17 datasets and Motion Capture datasets. Numbers correspond to the *S2T* metric $(\times 10^{-2})$.

| Freq Align | Feature Align | Weight | Aspirin | Benzene | Ethanol | Malonaldehyde | Naphthalene | Salicylic | Toluene | Uracil | Mocap-Walk | Mocap-Run |
|---|---|---|---|---|---|---|---|---|---|---|---|---|
| ✓ | ✓ | ✓ | 5.66 | 1.15 | 2.66 | 10.91 | 0.27 | 0.60 | 2.32 | 0.43 | 3.03 | 16.08 |
| ✓ | ✓ | - | 7.04 | 1.70 | 2.71 | 10.94 | 0.25 | 0.63 | 2.18 | 0.43 | 4.02 | 19.66 |

**Ablation on Truncation parameter $K$. (Table A.9)** Here, to compare performance with respect to the truncation parameter $K$, we trained and evaluated SGDD-EGNO across the MD17, motion capture, and protein datasets while varying $K$. Table A.9 reports the results on the motion capture dataset, Table A.10 presents the results on Aspirin—the molecule with the largest number of atoms in the MD17 dataset—and Table A.11 shows the results obtained on the protein dataset. Although performance varies with the choice of $K$, our goal was not to identify an optimal value but rather to

Table A.8: **Ablation study on weighting**. Results are reported for the SGDD-GFNODE model on the MD17 datasets and Motion Capture datasets. Numbers correspond to the *S2S* metric ($\times 10^{-2}$).

| Freq Align | Feature Align | Weight | Aspirin | Benzene | Ethanol | Malonaldehyde | Naphthalene | Salicylic | Toluene | Uracil | Mocap-Walk | Mocap-Run |
|---|---|---|---|---|---|---|---|---|---|---|---|---|
| ✓ | ✓ | ✓ | 7.38 | 2.23 | 3.64 | 12.72 | 0.34 | 0.79 | 4.97 | 0.52 | 5.82 | 34.46 |
| ✓ | ✓ | - | 8.46 | 3.50 | 3.61 | 12.73 | 0.32 | 0.80 | 4.70 | 0.53 | 7.71 | 36.77 |

select a reasonable $K$ based on the total number of available frequency components for each dataset (maximum $104$ for MD17, $155$ for Motion, and $3420$ for Protein). Accordingly, we used $K = 64$ for both MD17 and Motion, and $K = 128$ for Protein. Interestingly, experiments conducted with different values of $K$ show that values near $64$ consistently yield the best performance across all datasets. This aligns with the fact that long-term behavior in physical dynamics tasks is generally dominated by low-frequency components. Moreover, this observation provides a practical guideline: for new datasets or systems, initializing with $K \approx 64$ offers a robust starting point without requiring extensive tuning.

Table A.9: Comparison SGDD-EGNO on the motion capture dataset under different truncation level $K$ of the basis $B$. Upper part: *S2S*. Lower part: *S2T*.

| K | Walk | Run |
|---|---|---|
| 16 | 9.79±0.18 | 38.82±0.38 |
| 32 | 8.03±0.05 | 31.88±0.04 |
| 64 | 6.74±0.00 | 28.21±0.00 |
| 128 | 6.89±0.01 | 30.45±0.07 |
| full(155) | 7.26±0.02 | 28.18±0.18 |
| 16 | 4.7±0.10 | 17.99±1.39 |
| 32 | 3.96±0.06 | 15.37±1.16 |
| 64 | 3.22±0.05 | 13.52±0.89 |
| 128 | 3.29±0.05 | 14.39±1.05 |
| full(155) | 3.46±0.06 | 13.39±0.89 |

Table A.10: Comparison of SGDD-EGNO on MD17 (Aspirin) under different truncation levels $K$ of the basis $B$. Upper part: *S2S*. Lower part: *S2T*.

| K | Aspirin |
|---|---|
| 16 | 6.33 |
| 32 | 6.32 |
| 64 | 6.20 |
| full(104) | 6.26 |
| 16 | 8.04 |
| 32 | 8.06 |
| 64 | 7.84 |
| full(104) | 7.84 |

Table A.11: Comparison of SGDD-EGNO on the ADk equilibrium trajectory dataset under different truncation levels $K$ of the basis $B$ (S2S).

| K | ADk |
|---|---|
| 64 | 1.74 |
| 128 | 1.75 |
| 256 | 1.75 |
| 512 | 1.75 |

# E    ADDITIONAL VISUALIZATION

In addition, to demonstrate that the proposed SGDD framework achieves superior performance from a frequency perspective, we define a frequency loss and present the results in a table. The frequency loss is computed by mapping the predicted and ground-truth trajectories into the spectral domain via the spatio-temporal joint basis $B$, and then calculating the loss for each frequency component.

## E.1    MD17

**Comparison between SGDD-EGNO and EGNO**

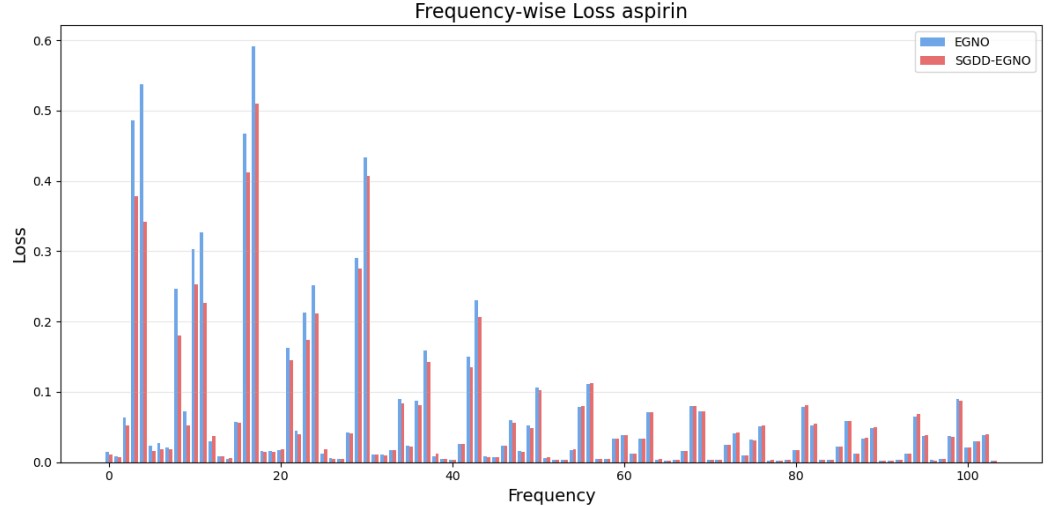

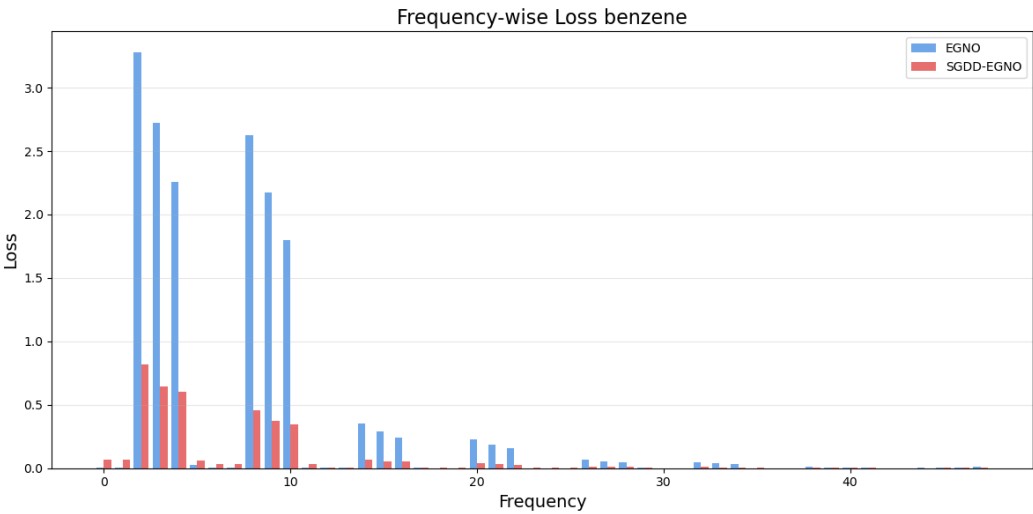

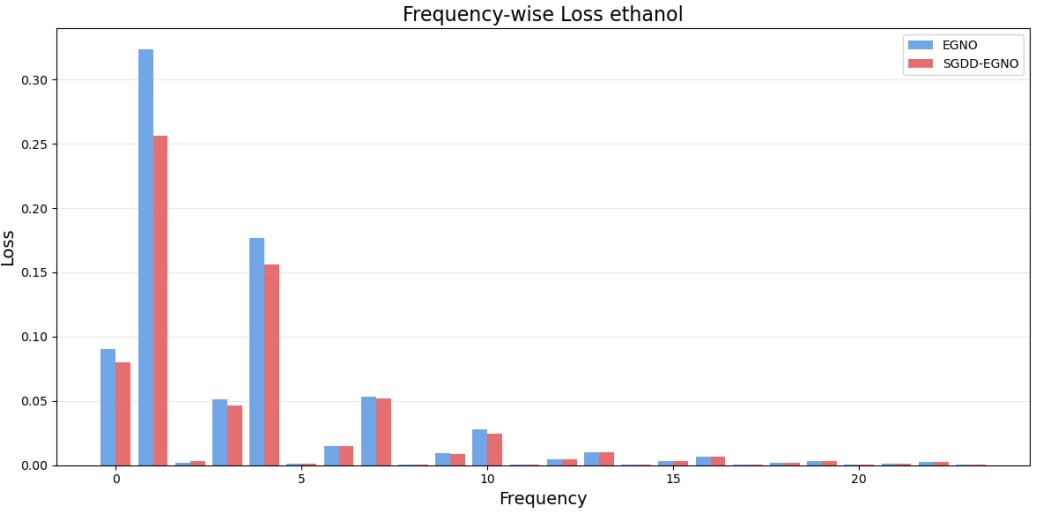

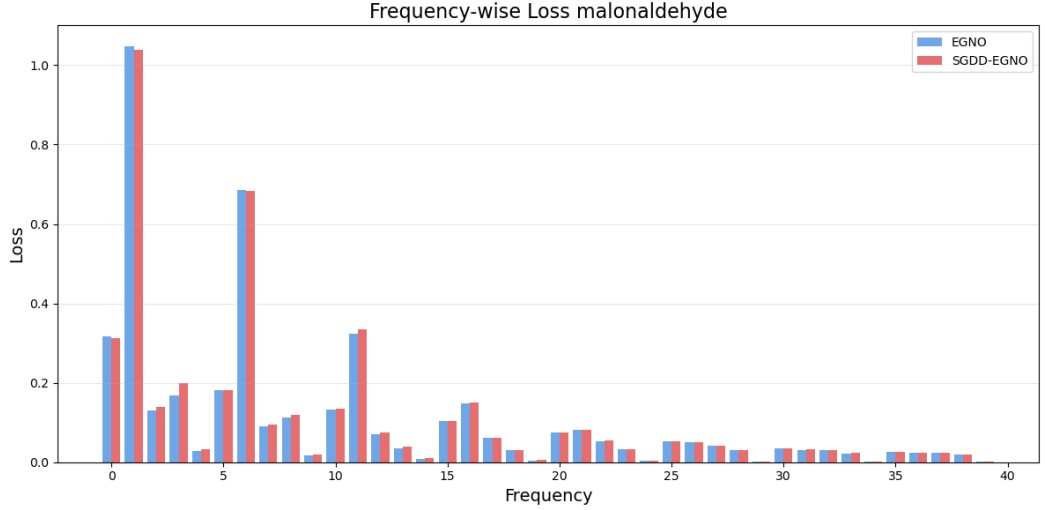

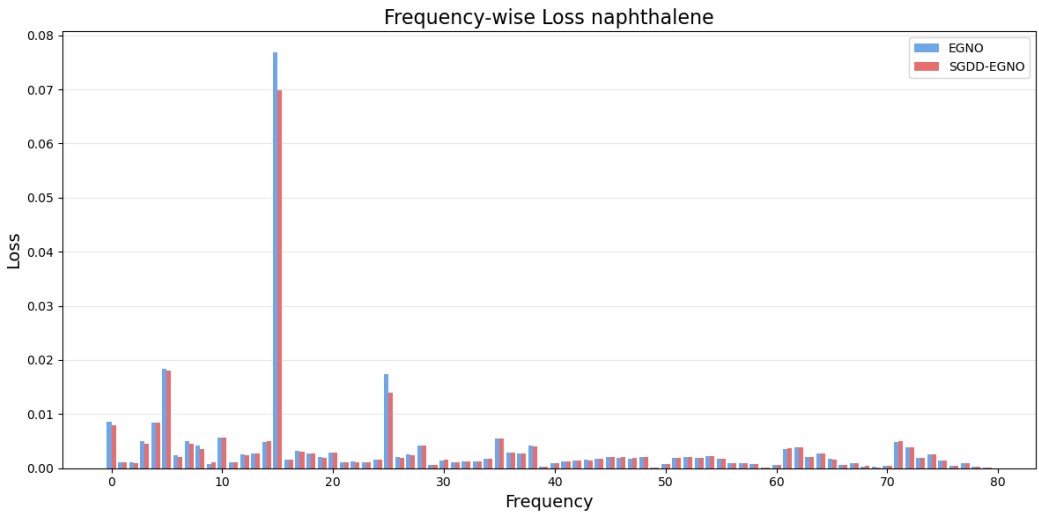

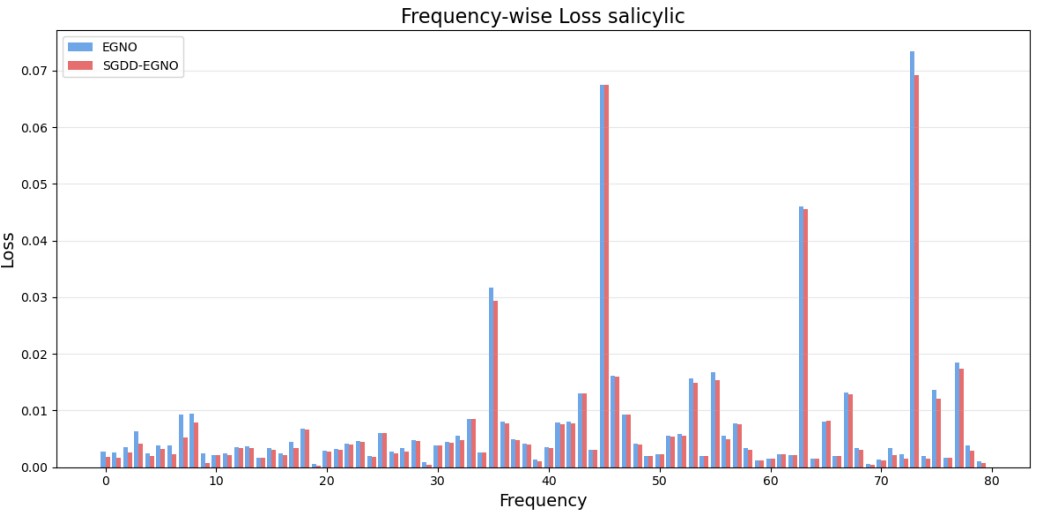

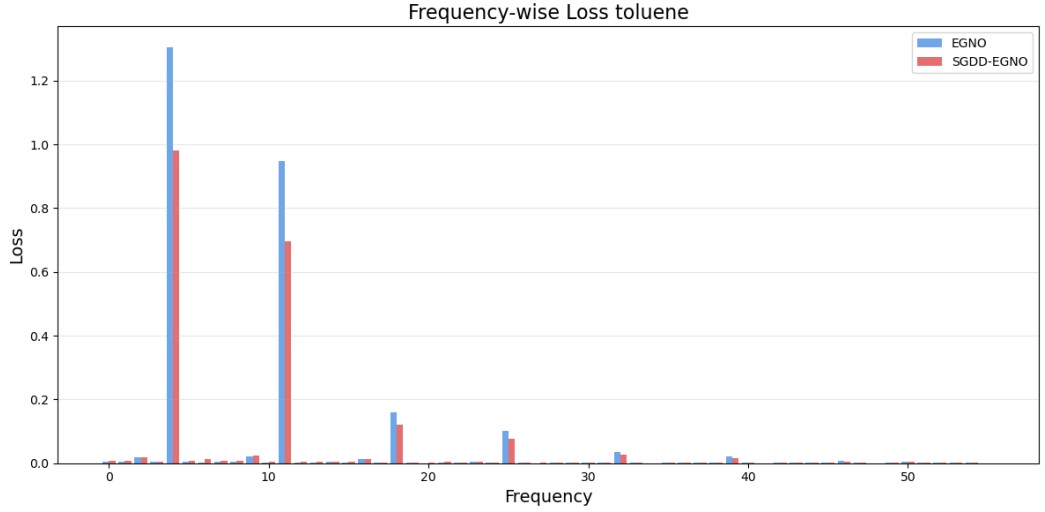

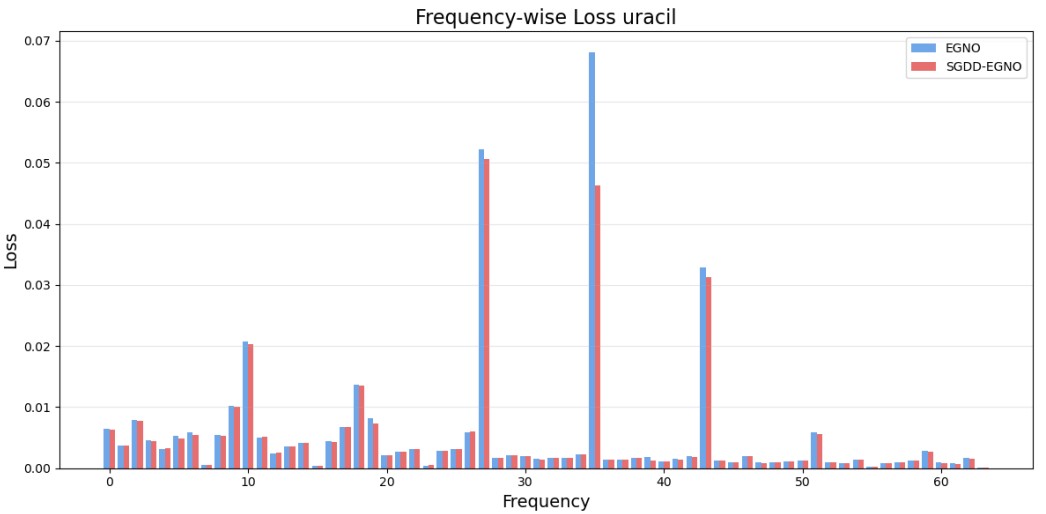

**Comparison between SGDD-GFNODE and GFNODE**

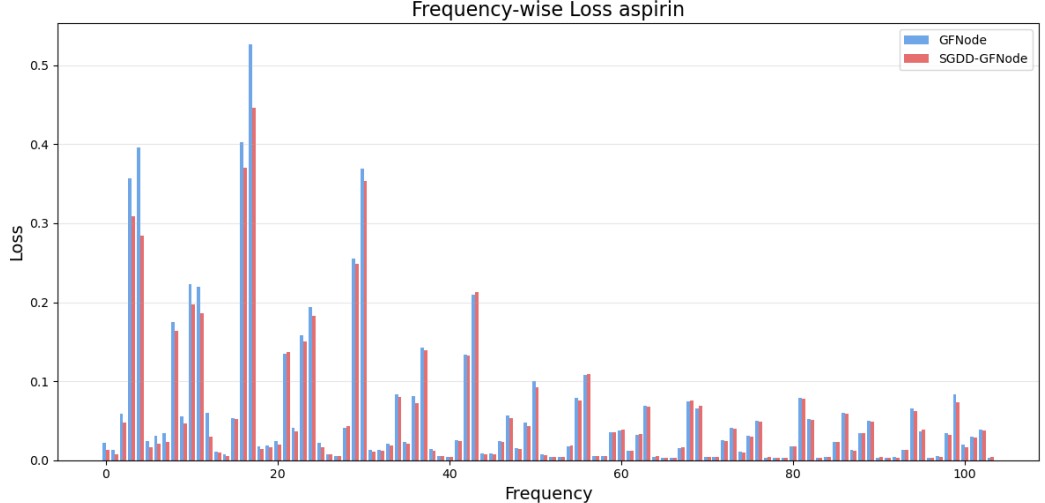

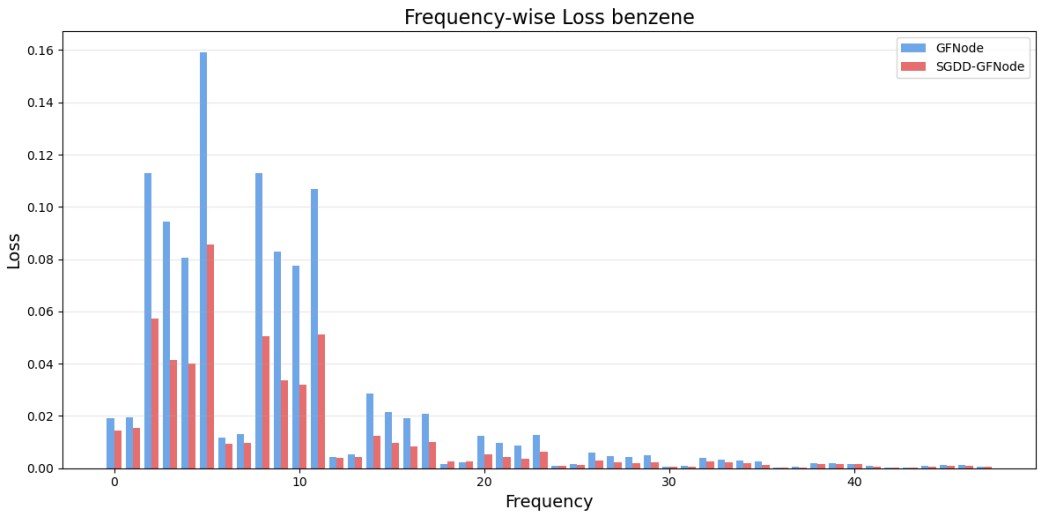

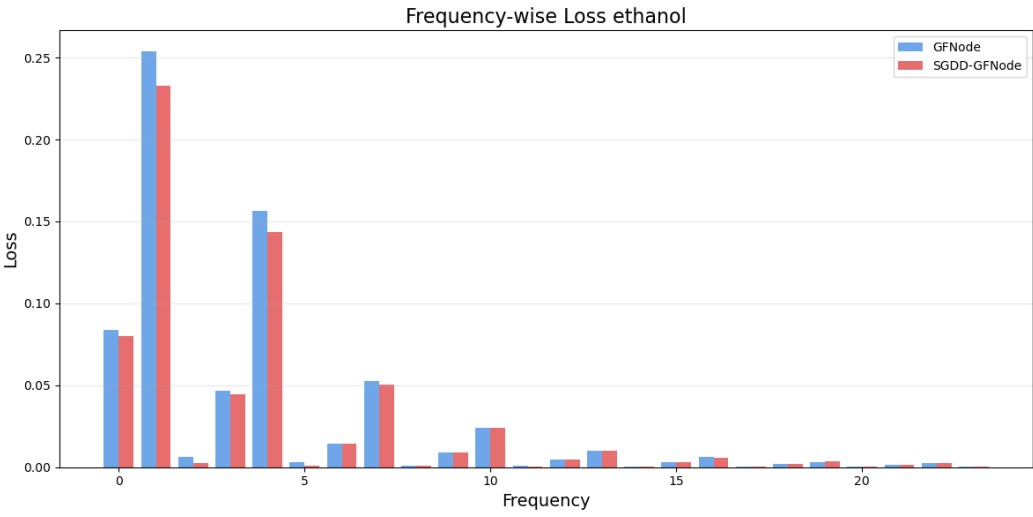

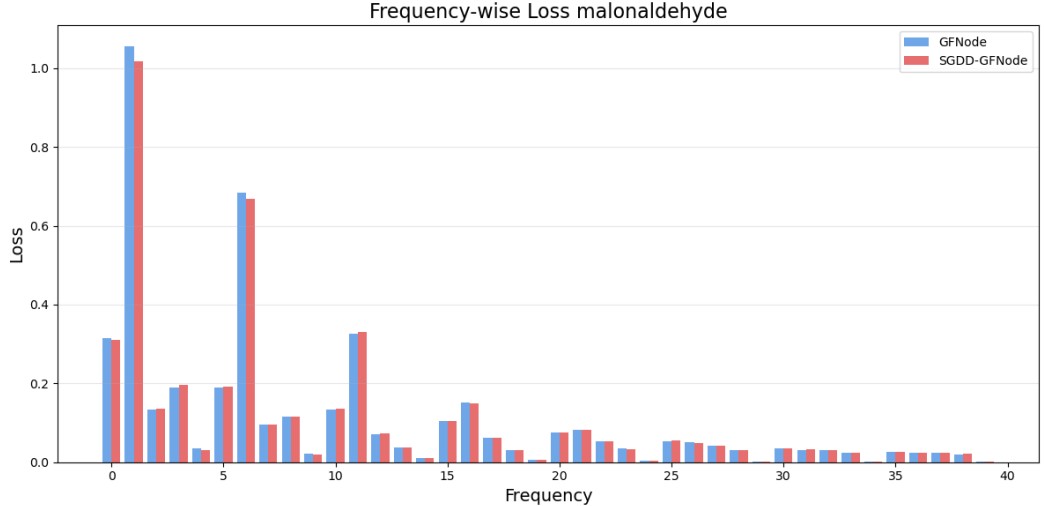

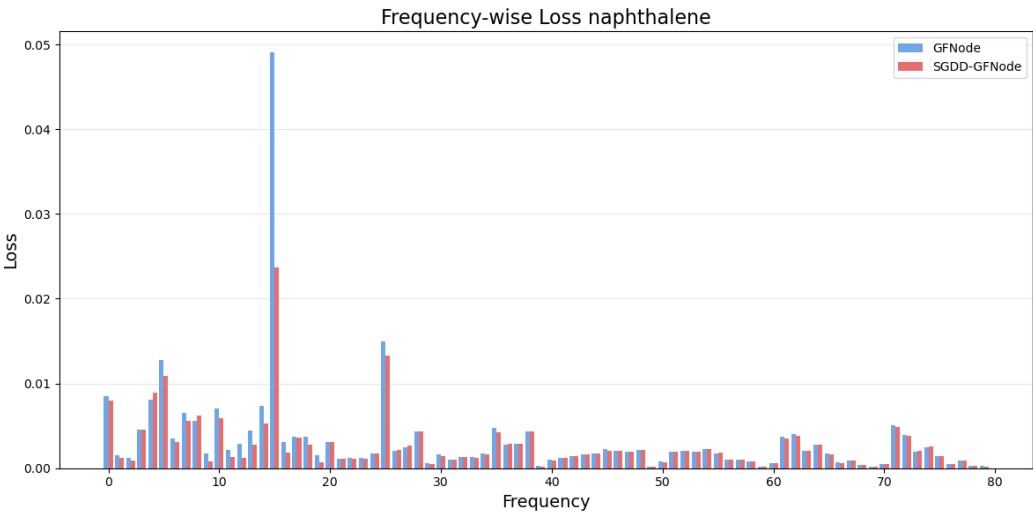

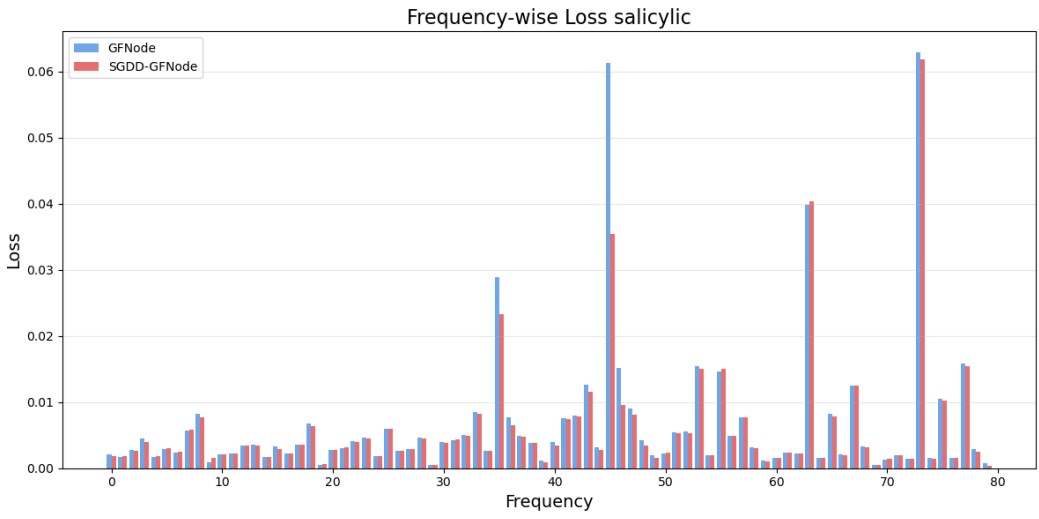

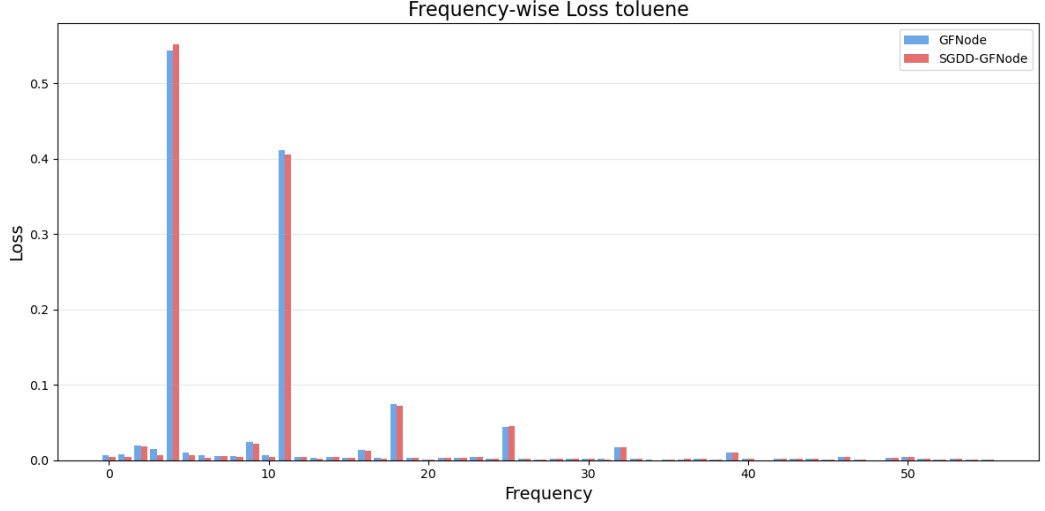

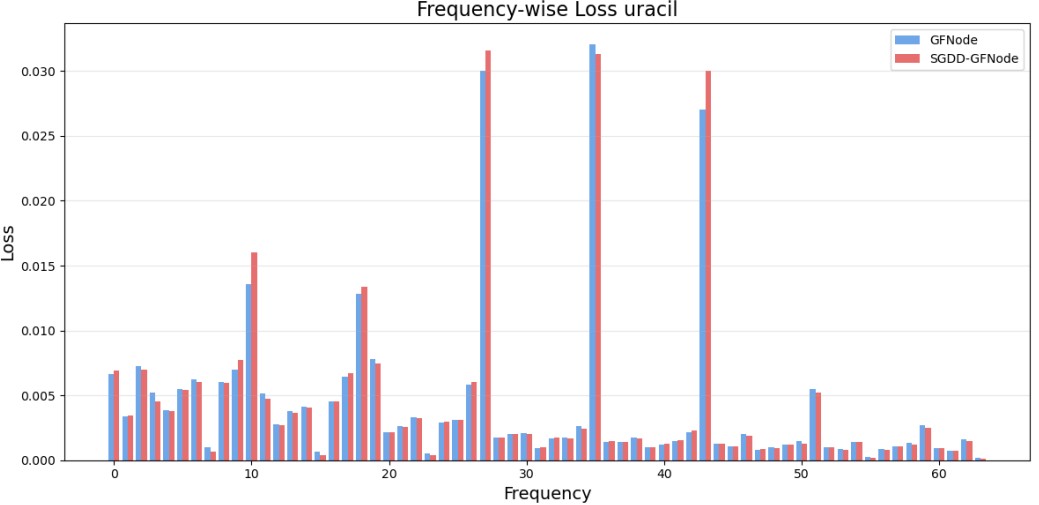

## E.2 MOTION CAPTURE.

**Comparison between SGDD-EGNO and EGNO**

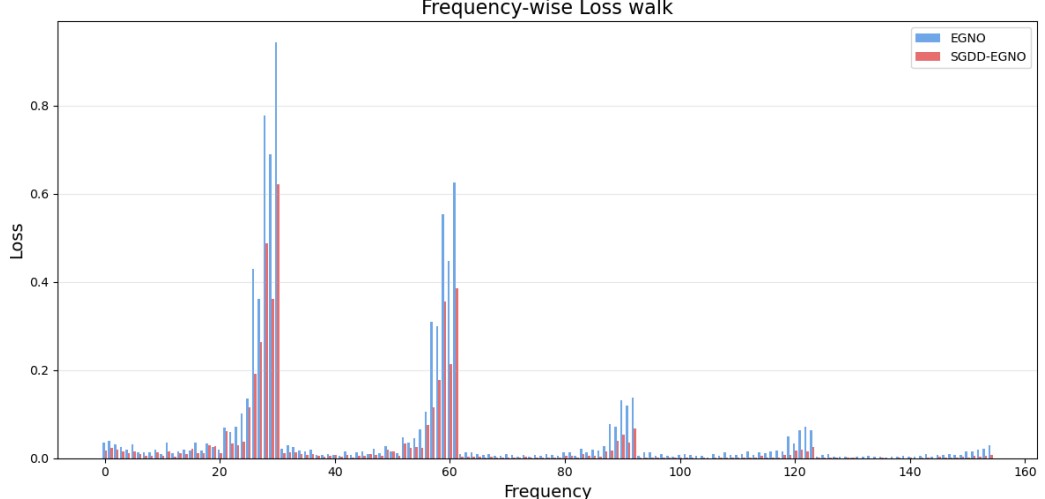

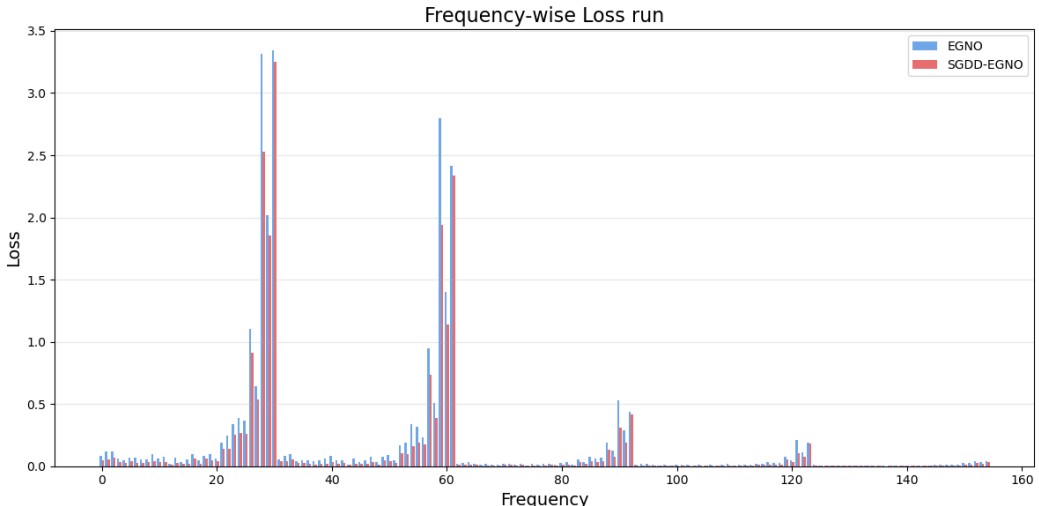

## Comparison between SGDD-GFNODE and GFNODE

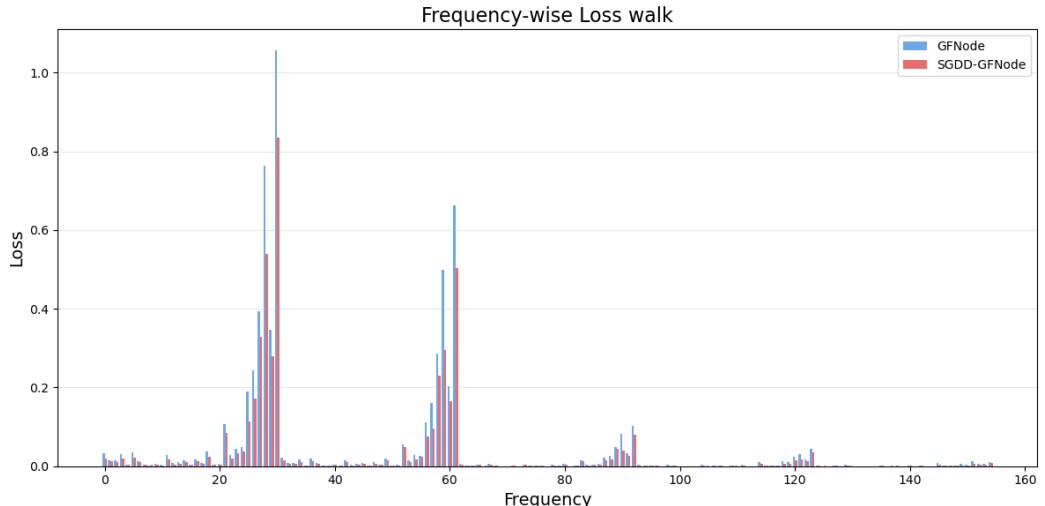

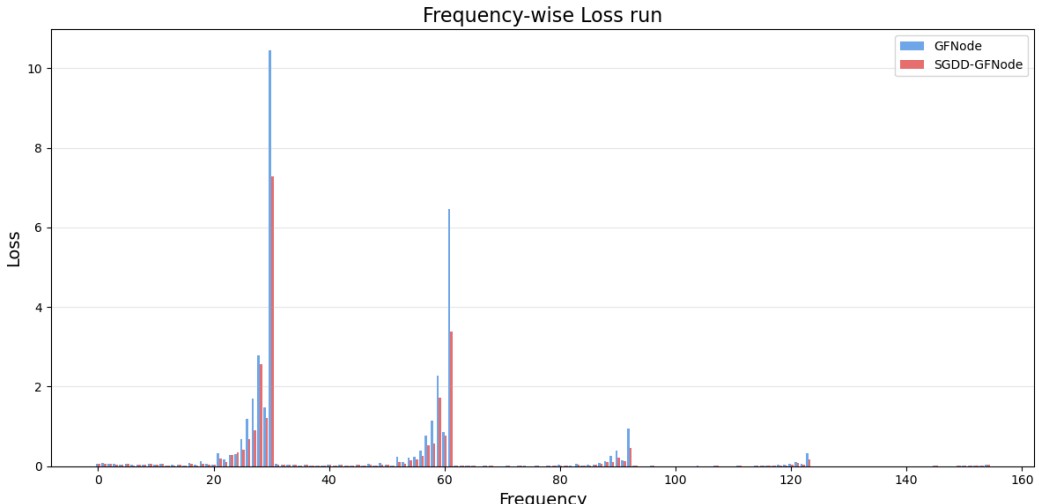

