# OpenReview forum: "Spectral-guided Physical Dynamics Distillation"
_ICLR.cc/2026/Conference — ICLR 2026 Poster_

### Official Review · Reviewer_6vVc · 2025-10-17

**Soundness:** 3
**Presentation:** 2
**Contribution:** 2
**Rating:** 4
**Confidence:** 3

**Summary:**

This work proposes Spectral-Guided Dynamics Distillation (SGDD) for long-horizon trajectory forecasting from only initial states. SGDD leverages full future trajectories as privileged information for a teacher encoder and distills that knowledge into a student encoder that must construct the full trajectory solely from the initial state. A spectral-guided enhancement module reweights joint spatiotemporal frequency components, enabling capture key low-frequency patterns.

**Strengths:**

- The paper is clearly written, well structured, and easy to follow.
- The joint handling of spatial and temporal information, particularly the emphasis on capturing overall low-frequency patterns, is insightful.
- The distillation strategy offers a practical way to train forecasting models that operate from initial conditions alone.
- Aligning the student with the teacher both in representation space and in the spectral domain is conceptually sound.

**Weaknesses:**

- Forecasting long-horizon time series **solely** from initial states is often unrealistic. In many real-world settings, future trajectories depend on external inputs, or may be influenced by events. For example, predicting a patient’s blood glucose purely from the current state ignores forthcoming meals (e.g., salad vs. burger). Because SGDD predicts future trajectories from only the initial state, it appears unable to handle such exogenously driven series. This is the main reason I lean toward a **weak reject**.
- As I understand it (see Question 2), the method relies on a spatiotemporal joint basis $B_{K} \in \mathbb{R}^{NT \times K}$. While this enables joint spatiotemporal modeling, it introduces two concerns:
  - Because the shape of $B_K$ explicitly depends on the number of time steps $T$, SGDD may only forecast a **fixed** number of steps at inference, limiting practical applicability.
  - For very high-dimensional systems or very long sequences, storing $B_{K}$ could become memory intensive.
- The primary experiments are in scientific domains, yet the method does not enforce physical constraints; the physical consistency of the forecasts is therefore uncertain.
- Experimental results should report mean ± standard deviation across multiple random seeds.
- If possible, evaluating SGDD with different encoders and decoders would better demonstrate the method’s robustness and generality.

**Questions:**

- I am a bit confused about the “spatial graph.” Each temporal graph $G_t$ already encodes the physical connections among particles. Why is an additional spatial graph $G_s$ mentioned?
- Do the authors build the joint basis $B_{K} \in \mathbb{R}^{NT \times K}$ from the training set and then **store** it for subsequent training and inference?
  - If **yes**, the storage and fixed-horizon issues in the Weakness section become significant.
  - If **no**, please clarify how $B_{K}$ is computed/updated and used in practice.
- Are the frequency-specific weights $w$ **different** for the teacher and student encoders? My understanding is **yes**.

---

> ### Author Response · Authors · 2025-11-21
>
> > **W1. Forecasting solely from initial states**
>
> **A:** We sincerely thank the reviewer for raising this important point. We fully agree that **in many real-world scenarios—such as glucose prediction influenced by meals—future trajectories depend on external inputs**, and forecasting solely from an initial state would indeed be inappropriate. However, the settings considered in this work fall into a different class of problems. Our focus is on **closed and self-contained physical systems**, such as molecular dynamics, protein equilibrium trajectories, and human motion from a given initial pose, where the temporal evolution is governed primarily by internal physical interactions rather than exogenous events.
>
> This modeling choice is consistent with the design of the baseline decoders used in our work (e.g., EGNO, GFNode), which also **assume deterministic evolution from an initial state and do not incorporate external covariates**. Across these domains, such an assumption is standard: MD17 molecules evolve according to Newtonian dynamics under fixed potential energy surfaces; protein equilibrium simulations follow physically constrained stochastic dynamics; and human motion rollouts from a known starting configuration are often treated as autonomous trajectories. In these settings, **initial-state forecasting is a standard and meaningful evaluation protocol**, particularly for assessing long-horizon stability, which is the primary challenge we aim to address.
>
> We agree that **extending SGDD to systems with external forcing or control inputs**—such as exogenous events, interventions, or conditional signals—would broaden its applicability. We view this as a valuable direction for future work, and we appreciate the reviewer for highlighting this distinction. Nonetheless, we hope the clarification above helps contextualize the assumptions underlying our current problem formulation and the relevance of SGDD to the physical domains studied in the paper.
>
> > **W2. Spatiotemporal joint basis: fixed horizon & memory concerns**
>
> **A:** We appreciate the reviewer’s thoughtful questions regarding the implications of using a spatiotemporal joint basis. **In our setting—long-horizon physical dynamics forecasting—it is standard to use the same prediction horizon for both training and inference**, and SGDD follows this conventional practice. This is also true for the baseline decoders used in our work: **both EGNO and GFNode are trained and evaluated under a fixed length**, and their architectures do not support variable-length prediction without retraining. **In this sense, constructing the joint basis for a specified horizon does not impose any additional limitation** beyond what is already inherent in existing fixed-horizon forecasting frameworks.
>
> That said, we acknowledge the reviewer’s broader point. **Designing architectures that can flexibly handle variable-length or dynamically changing horizons—without reconstructing the basis—would be an interesting direction for future work**, and we see this as a promising avenue to explore.
>
> Regarding the memory cost of storing the spatiotemporal joint basis, **we note that the basis is constructed once per system because our spatio-temporal graph is time-invariant**. Therefore, the basis does not grow with each rollout and is not recomputed across time steps. **Even for higher-dimensional systems, this results in a single set of eigenvectors rather than a time-dependent collection**, keeping the memory footprint manageable in practice. Moreover, **only the truncated portion of the basis is used during alignment**, further reducing storage requirements.

---

> ### Author Response · Authors · 2025-11-21
>
> > **W3. Experimental results should report mean and standard deviation across multiple random seeds**
>
> **A:** We thank you for pointing out the lack of statistical reporting in the earlier version. Following this suggestion, we have updated the main experiments on the MD17 dataset (Table 1) by running all SGDD models with three independent seeds and reporting mean and variance. We hope this provides a clearer picture of the robustness of our method.
>
> ### **Table 1. MSE (×10−2) on MD17 dataset.**
> | **S2S** |      | Aspirin | Benzene | Ethanol | Malonaldehyde | Naphthalene | Salicylic | Toluene | Uracil |
> |--------|------|---------|---------|---------|----------------|-------------|-----------|---------|--------|
> | RF | | 10.94±0.01 | 103.72±1.29 | 4.64±0.01 | 13.93±0.03 | 0.50±0.01 | 1.23±0.01 | 10.93±0.04 | 0.64±0.01 |
> | TFN | | 12.37±0.18 | 58.48±1.98 | 4.81±0.04 | 13.62±0.08 | 0.49±0.01 | 1.03±0.02 | 10.89±0.04 | 0.84±0.02 |
> | SE(3)-Tr. | | 11.12±0.06 | 68.11±0.67 | 4.74±0.13 | 13.89±0.02 | 0.52±0.01 | 1.13±0.02 | 10.88±0.06 | 0.79±0.02 |
> | EGNN | | 14.41±0.15 | 62.40±0.53 | 4.64±0.01 | 13.64±0.01 | 0.47±0.02 | 1.02±0.02 | 11.78±0.07 | 0.64±0.01 |
> | EGNN-R | | 14.51±0.19 | 62.61±0.75 | 4.94±0.21 | 17.25±0.05 | 0.82±0.02 | 1.35±0.02 | 11.59±0.04 | 1.11±0.02 |
> | EGNN-S | | 9.50±0.10 | 66.45±0.89 | 4.63±0.01 | 12.88±0.01 | 0.45±0.01 | 1.00±0.02 | 10.78±0.05 | 0.60±0.01 |
> | EGNO | | 9.18±0.06 | 48.85±0.55 | 4.62±0.01 | _12.80±0.02_ | 0.37±0.01 | 0.86±0.02 | 10.21±0.05 | **0.52±0.02** |
> | GFNODE | | 7.93±0.00 | _4.82±0.00_ | _3.92±0.00_ | 12.87±0.00 | 0.37±0.00 | _0.80±0.00_ | **4.82±0.00** | 0.54±0.00 |
> | **SGDD-EGNO** | | _7.84±0.00_ | 12.97±0.00 | 4.04±0.00 | 12.98±0.00 | _0.36±0.00_ | 0.85±0.00 | 9.45±0.01 | _0.53±0.00_ |
> |  | | (+14.5%) | (+73.4%) | (+12.5%) | (-1.4%) | (+2.7%) | (+1.1%) | (+7.4%) | (-1.9%) |
> | **SGDD-GFNODE** | | **7.29±0.00** | **2.74±0.00** | **3.64±0.00** | **12.72±0.00** | **0.33±0.00** | **0.79±0.00** | _5.16±0.00_ | _0.53±0.00_ |
> |  | | (+8.1%) | (+43.2%) | (+7.1%) | (+1.2%) | (+10.8%) | (+1.3%) | (-7.1%) | (+1.9%) |
>
> | **S2T** |      |         |         |         |              |             |           |         |        |
> |--------|------|---------|---------|---------|--------------|-------------|-----------|---------|--------|
> | NDCN | | 31.73±0.40 | 56.21±0.30 | 10.74±0.02 | 46.55±0.28 | 2.25±0.01 | 3.58±0.11 | 13.92±0.02 | 2.38±0.00 |
> | ITO | | 20.56±0.03 | 57.25±0.58 | 8.60±0.27 | 28.44±0.73 | 1.82±0.17 | 2.48±0.34 | 12.47±0.30 | 1.33±0.12 |
> | LG-ODE | | 19.36±0.12 | 53.92±1.32 | 7.08±0.01 | 24.41±0.03 | 1.73±0.02 | 3.82±0.04 | 11.18±0.04 | 2.11±0.02 |
> | EGNN | | 9.24±0.07 | 57.85±2.70 | 4.63±0.00 | 12.81±0.01 | 0.38±0.01 | 0.85±0.00 | 10.41±0.04 | 0.56±0.02 |
> | EGNN-R | | 12.07±0.11 | 23.73±0.30 | 3.44±0.17 | 13.38±0.03 | 0.63±0.01 | 1.15±0.02 | 5.04±0.02 | 0.89±0.01 |
> | EGNN-S | | 9.49±0.12 | 29.99±0.65 | 3.29±0.01 | 11.21±0.01 | 0.43±0.01 | 1.36±0.02 | 4.85±0.04 | 0.68±0.01 |
> | EGNO | | 7.37±0.07 | 22.41±0.31 | 3.28±0.02 | _10.67±0.01_ | 0.32±0.01 | 0.77±0.01 | 4.58±0.03 | 0.47±0.01 |
> | GFNODE | | _6.07±0.09_ | _1.51±0.07_ | _2.74±0.01_ | **9.43±0.02** | **0.24±0.02** | _0.63±0.05_ | **1.80±0.03** | **0.41±0.02** |
> | **SGDD-EGNO** | | 6.20±0.01 | 7.79±0.18 | 2.88±0.01 | 11.01±0.05 | 0.33±0.00 | 0.69±0.00 | 4.23±0.09 | 0.50±0.00 |
> |  | | (+15.8%) | (+65.2%) | (+12.1%) | (-3.1%) | (-3.1%) | (+10.3%) | (+7.6%) | (-6.3%) |
> | **SGDD-GFNODE** | | **5.63±0.01** | **1.36±0.01** | **2.67±0.01** | 10.95±0.04 | _0.26±0.00_ | **0.60±0.00** | _2.39±0.03_ | _0.44±0.00_ |
> |  | | (+7.2%) | (+9.9%) | (+2.5%) | (-16.1%) | (-8.3%) | (+4.7%) | (-28.8%) | (-4.8%) |

---

> ### Author Response · Authors · 2025-11-21
>
> > **W4. Evaluating SGDD with different encoders and decoders**
>
> **A:** We thank the reviewer for highlighting the importance of assessing SGDD across different encoder–decoder choices to better demonstrate the robustness and generality of the framework. In the original submission, SGDD employed EGNO and GFNode as decoders, STSGNN as the dynamics encoder, and GAT as the initial encoder. We agree that this set of encoders was limited. To address this concern, we expanded our analysis by implementing additional encoder variants---STGCN as an alternative dynamics encoder and GINE or Transformer as alternative initial encoders---and evaluating all combinations under identical training settings.
>
> The new ablation results (Table 5) show that changes to $E_{\text{init}}$ induce larger performance variation than changes to $E_{\text{dyn}}$.
> GINE yields performance comparable to GAT, whereas Transformer exhibits degradation on several molecules due to its substantially larger parameter count, which leads to overfitting or underfitting under the same optimization configuration.
>
> These findings indicate that SGDD is moderately robust to reasonable encoder substitutions (e.g., GAT $\leftrightarrow$ GINE or STSGNN $\leftrightarrow$ STGCN), while heavier encoders may require different training strategies to realize their potential.
>
> Overall, this analysis demonstrates that SGDD remains effective across a range of encoder architectures, supporting its robustness and generality beyond a specific model choice. For further details, please refer to the newly added ablation analysis section 4.4.3 in the manuscript.
>
> ### **Table 5. MSE (×10⁻²) of SGDD-EGNO across different encoder combinations on MD17 (S2S)**
>
> | **$E_{\text{init}}$** | **$E_{\text{dyn}}$** | **Aspirin** | **Ethanol** | **Naphthalene** | **Salicylic** | **Uracil** |
> |------------------------|------------------------|-------------|-------------|------------------|----------------|-------------|
> | GAT (142K)            | STSGNN (397K)          | 7.75        | 4.00        | 0.36             | 0.85           | 0.52        |
> | GAT (142K)            | STGCN (151K)           | 7.97        | 4.08        | 0.36             | 0.84           | 0.52        |
> | GINE (336K)           | STSGNN (397K)          | 7.85        | 4.21        | 0.36             | 0.86           | 0.53        |
> | GINE (336K)           | STGCN (151K)           | 8.02        | 4.15        | 0.36             | 0.86           | 0.52        |
> | Transformer (541K)    | STSGNN (397K)          | 8.90        | 4.12        | 0.36             | 0.91           | 0.53        |
> | Transformer (541K)    | STGCN (151K)           | 8.69        | 4.27        | 0.36             | 0.94           | 0.53        |
>
> > **Q1. Confusion about the notion**
>
> **A:** We appreciate the reviewer’s comment and apologize for the confusion caused by our notation.
> In the original version, the symbol $\mathcal{G}\_t$ was used both for the state graph at time $t$ (which contains the spatial connectivity and state information) and for the temporal graph used to define the temporal basis.
> This overlap understandably led to ambiguity, so we have now renamed the temporal graph as $\mathcal{G}_{\text{temp}}$.
>
> The temporal graph $\mathcal{G}_{\{temp}}$ is a simple chain structure over the time indices: each node corresponds to a time step, and edges connect adjacent time steps to represent temporal continuity as below. This temporal-graph construction has now been clarified in Appendix A.2. We hope this clarification resolves the confusion.
>
>
> #### **Temporal graph**
>
> The temporal graph is:
>
> $$
> \mathcal{G}_{temp} = (V_t, E_t)
> $$
>
> with:
>
> $$
> V_t = \{1, \dots, T\}
> $$
>
> Temporal continuity is modeled using a 1D chain:
>
> $$
> E_t = \{(i, i+1) \mid 1 \le i < T\}
> $$

---

> ### Author Response · Authors · 2025-11-21
>
> > **Q2. Joint basis construction & storage**
>
> **A.**  We sincerely thank the reviewer for the question and the opportunity to clarify this point. The **spatiotemporal joint basis is computed once per system**, using the time-invariant spatial graph together with the simple temporal chain graph. **A single basis is shared across the entire trajectory**, and it is **not constructed separately** for individual training samples or rollouts.
>
> As discussed earlier, the **fixed-horizon and memory-cost concerns do not arise in practice** under our setting. Long-horizon physical dynamics forecasting conventionally uses the same prediction horizon for both training and inference, and the baseline decoders we build upon (EGNO and GFNode) also operate under fixed horizons without supporting variable-length prediction. Thus, **defining the joint basis for a specified horizon does not impose any additional restriction** beyond standard practice.
>
> That said, we fully acknowledge the reviewer’s broader point, and **developing architectures capable of handling variable-length or dynamically changing horizons** would indeed be an interesting and valuable direction for future work.
>
> > **Q3. Frequency-specific weights for teacher vs. student**
>
> **A.**  We thank the reviewer for the question. The frequency-specific weights are not separately learned for the teacher and the student. Instead, a single set of learnable spectral weights is used and shared during alignment. This design ensures that the teacher’s spectral structure serves as a stable reference, and the student is aligned to the same frequency scaling rather than learning an independent weighting scheme.

---

> ### Comment · Reviewer_6vVc · 2025-11-27
>
> Thank you for your thorough responses. My concerns have been addressed. I’m pleased to raise my score to 6. Wishing you a happy Thanksgiving!

---

> > ### Author Response · Authors · 2025-11-28
> > **Thank you very much for your thoughtful assessment and for raising your score.**
> >
> > Thank you very much for your thoughtful assessment and for raising your score. We truly appreciate the time and care you invested throughout the review and discussion phases. We are glad that our clarifications addressed your concerns, and we are grateful for the constructive insights you provided — they genuinely helped us strengthen the paper and also inspired valuable future directions.
> > Wishing you a happy Thanksgiving as well!

---

### Official Review · Reviewer_Podc · 2025-10-28

**Soundness:** 3
**Presentation:** 3
**Contribution:** 3
**Rating:** 6
**Confidence:** 3

**Summary:**

The paper proposes SGDD, a “spectral-guided dynamics distillation” framework for long-horizon particle-dynamics forecasting that trains a teacher encoder on full future trajectories as privileged information and distills its spatio-temporal representation into a student that only sees the initial state at test time. Concretely, both teacher and student representations are mapped into a truncated joint spatio-temporal spectral basis built from temporal and spatial Laplacians. A learnable frequency-weighting module emphasizes low-frequency modes while preserving residual components, and training aligns the student and teacher both in representation space and in the spectral coefficients.

**Strengths:**

1. The joint spatio-temporal spectral basis with learnable per-mode weights provides a principled way to emphasize low-frequency global trends while preserving high-frequency residuals, which directly targets long-horizon stability.
2. The dual alignment together with gradient detaching for the teacher is a clean instantiation of privileged-information distillation that avoids trivial position-level imitation and back-door leakage during training.
3. On MD17, the method demonstrates large, interpretable gains specifically where baseline errors concentrate in low frequencies.
4. Ablations indicate that both spectral- and feature-space alignments contribute, and that the truncation K trades off emphasis breadth against noise, which is informative for practitioners.

**Weaknesses:**

1. Statistical rigor is uneven, with several results reported as single numbers or with “±0.0” and no confidence intervals, and with seeds unspecified for some tables, which limits conclusions about robustness.
2. Fairness and tuning budgets are not fully documented. This is important because SGDD wraps baseline decoders; strong gains might partly arise from extra representation capacity or training length unless search spaces and early-stopping criteria are matched and reported.
3. The method’s dependence on decoder capacity is acknowledged but not quantified.

**Questions:**

1. Could you replicate all baselines within your codebase with matched search to rule out confounds from external reported numbers, and add a “decoder-capacity-matched” comparison where the decoder architecture and training schedule are identical with and without SGDD?
2. The frequency weighting advocates prioritizing low-frequency modes. Can you provide frequency-conditioned error profiles across datasets (as in Fig. 4 for Benzene) and show at least one regime where high-frequency content is crucial? Do gains persist there, or does performance degrade as K grows?
3. Beyond qualitative statements, it would be good to quantify decoder dependence by swapping in a weaker and a stronger decoder and plotting accuracy vs. decoder capacity to show that SGDD’s benefit is not merely a proxy for more parameters.

---

> ### Author Response · Authors · 2025-11-21
>
> > **W1. Statistical rigor**
>
> **A.** We thank the reviewer for pointing out the lack of statistical reporting in the earlier version. Following this suggestion, we have updated the main experiments on the MD17 dataset by running all SGDD models with three independent seeds and reporting the mean and variance in Table 1. We hope this provides a clearer picture of the robustness of our method.
>
> Regarding entries that appear as “$\pm 0.00$”, these correspond to extremely small variances that round to zero under our decimal formatting.
>
> In **Table 1 (MD17)**, the overall variance across methods — including both our model and prior baselines — is already very small.
> In **Table 2 (Motion dataset)**, although some baselines show higher variance, the differences in mean performance remain sufficiently large that the relative ranking is unaffected.
>
> We hope these explanations clarify the statistical reporting in our experiments, and we sincerely appreciate the reviewer’s attention to this aspect of the evaluation.
>
> ### **Table 1. MSE (×10−2) on MD17 dataset.**
> | **S2S** |      | Aspirin | Benzene | Ethanol | Malonaldehyde | Naphthalene | Salicylic | Toluene | Uracil |
> |--------|------|---------|---------|---------|----------------|-------------|-----------|---------|--------|
> | RF | | 10.94±0.01 | 103.72±1.29 | 4.64±0.01 | 13.93±0.03 | 0.50±0.01 | 1.23±0.01 | 10.93±0.04 | 0.64±0.01 |
> | TFN | | 12.37±0.18 | 58.48±1.98 | 4.81±0.04 | 13.62±0.08 | 0.49±0.01 | 1.03±0.02 | 10.89±0.04 | 0.84±0.02 |
> | SE(3)-Tr. | | 11.12±0.06 | 68.11±0.67 | 4.74±0.13 | 13.89±0.02 | 0.52±0.01 | 1.13±0.02 | 10.88±0.06 | 0.79±0.02 |
> | EGNN | | 14.41±0.15 | 62.40±0.53 | 4.64±0.01 | 13.64±0.01 | 0.47±0.02 | 1.02±0.02 | 11.78±0.07 | 0.64±0.01 |
> | EGNN-R | | 14.51±0.19 | 62.61±0.75 | 4.94±0.21 | 17.25±0.05 | 0.82±0.02 | 1.35±0.02 | 11.59±0.04 | 1.11±0.02 |
> | EGNN-S | | 9.50±0.10 | 66.45±0.89 | 4.63±0.01 | 12.88±0.01 | 0.45±0.01 | 1.00±0.02 | 10.78±0.05 | 0.60±0.01 |
> | EGNO | | 9.18±0.06 | 48.85±0.55 | 4.62±0.01 | _12.80±0.02_ | 0.37±0.01 | 0.86±0.02 | 10.21±0.05 | **0.52±0.02** |
> | GFNODE | | 7.93±0.00 | _4.82±0.00_ | _3.92±0.00_ | 12.87±0.00 | 0.37±0.00 | _0.80±0.00_ | **4.82±0.00** | 0.54±0.00 |
> | **SGDD-EGNO** | | _7.84±0.00_ | 12.97±0.00 | 4.04±0.00 | 12.98±0.00 | _0.36±0.00_ | 0.85±0.00 | 9.45±0.01 | _0.53±0.00_ |
> |  | | (+14.5%) | (+73.4%) | (+12.5%) | (-1.4%) | (+2.7%) | (+1.1%) | (+7.4%) | (-1.9%) |
> | **SGDD-GFNODE** | | **7.29±0.00** | **2.74±0.00** | **3.64±0.00** | **12.72±0.00** | **0.33±0.00** | **0.79±0.00** | _5.16±0.00_ | _0.53±0.00_ |
> |  | | (+8.1%) | (+43.2%) | (+7.1%) | (+1.2%) | (+10.8%) | (+1.3%) | (-7.1%) | (+1.9%) |
>
> | **S2T** |      |         |         |         |              |             |           |         |        |
> |--------|------|---------|---------|---------|--------------|-------------|-----------|---------|--------|
> | NDCN | | 31.73±0.40 | 56.21±0.30 | 10.74±0.02 | 46.55±0.28 | 2.25±0.01 | 3.58±0.11 | 13.92±0.02 | 2.38±0.00 |
> | ITO | | 20.56±0.03 | 57.25±0.58 | 8.60±0.27 | 28.44±0.73 | 1.82±0.17 | 2.48±0.34 | 12.47±0.30 | 1.33±0.12 |
> | LG-ODE | | 19.36±0.12 | 53.92±1.32 | 7.08±0.01 | 24.41±0.03 | 1.73±0.02 | 3.82±0.04 | 11.18±0.04 | 2.11±0.02 |
> | EGNN | | 9.24±0.07 | 57.85±2.70 | 4.63±0.00 | 12.81±0.01 | 0.38±0.01 | 0.85±0.00 | 10.41±0.04 | 0.56±0.02 |
> | EGNN-R | | 12.07±0.11 | 23.73±0.30 | 3.44±0.17 | 13.38±0.03 | 0.63±0.01 | 1.15±0.02 | 5.04±0.02 | 0.89±0.01 |
> | EGNN-S | | 9.49±0.12 | 29.99±0.65 | 3.29±0.01 | 11.21±0.01 | 0.43±0.01 | 1.36±0.02 | 4.85±0.04 | 0.68±0.01 |
> | EGNO | | 7.37±0.07 | 22.41±0.31 | 3.28±0.02 | _10.67±0.01_ | 0.32±0.01 | 0.77±0.01 | 4.58±0.03 | 0.47±0.01 |
> | GFNODE | | _6.07±0.09_ | _1.51±0.07_ | _2.74±0.01_ | **9.43±0.02** | **0.24±0.02** | _0.63±0.05_ | **1.80±0.03** | **0.41±0.02** |
> | **SGDD-EGNO** | | 6.20±0.01 | 7.79±0.18 | 2.88±0.01 | 11.01±0.05 | 0.33±0.00 | 0.69±0.00 | 4.23±0.09 | 0.50±0.00 |
> |  | | (+15.8%) | (+65.2%) | (+12.1%) | (-3.1%) | (-3.1%) | (+10.3%) | (+7.6%) | (-6.3%) |
> | **SGDD-GFNODE** | | **5.63±0.01** | **1.36±0.01** | **2.67±0.01** | 10.95±0.04 | _0.26±0.00_ | **0.60±0.00** | _2.39±0.03_ | _0.44±0.00_ |
> |  | | (+7.2%) | (+9.9%) | (+2.5%) | (-16.1%) | (-8.3%) | (+4.7%) | (-28.8%) | (-4.8%) |

---

> ### Author Response · Authors · 2025-11-21
>
> > **Q1 & Q3 (& W3). Baseline replication & Decoder-capacity dependence**
>
> **A:** We appreciate the reviewer’s thoughtful questions regarding **fairness**, **decoder capacity**, and the possibility that **SGDD’s improvements may partially arise from differences in model size or training budgets**. We would like to note that we provide the performance of our independently implemented **EGNO and GFNode** in Tables A.3 and A.4. Although these implementations are written within our codebase, they are designed to follow the official implementations as closely as possible. Importantly, **the EGNO and GFNode decoders used within SGDD are architecturally identical to these standalone versions**. Thus, **there is no difference in capacity** between the decoders used in **SGDD-EGNO / SGDD-GFNode** and the corresponding standalone EGNO / GFNode baselines.
>
> We also understand the reviewer’s concern that **differences in training schedules** could introduce additional confounding factors. The standalone EGNO and GFNode baselines are trained using the best-performing schedules recommended by the original papers, whereas SGDD-EGNO and SGDD-GFNode follow the SGDD training procedure. To examine whether this discrepancy could influence the reported gains, **we additionally trained standalone EGNO on the MD17 dataset using the SGDD training schedule**. The only differences between the two schedules are the learning rate choice and the use of a scheduler. In Table below, “EGNO (implemented)” refers to the model trained with EGNO’s original schedule, and “EGNO (SGDD setting)” refers to the same model trained under the SGDD schedule. **The results show that EGNO (SGDD setting) performs nearly the same as EGNO (implemented)**, suggesting that changing the training schedule does not meaningfully alter the performance of the baseline. Based on these observations, **we believe the improvements observed in SGDD-EGNO are attributable to the proposed representation-learning framework rather than differences in decoder capacity or training configurations**.
>
> We hope this clarification addresses the reviewer’s concerns, and we thank the reviewer again for raising this important point.
>
> ---
>
> ### Table A.3. Comparison on the MD17 dataset with our own implementations
>
>
>
> | S2S | Aspirin | Benzene | Ethanol | Malonaldehyde | Naphthalene | Salicylic | Toluene | Uracil |
> |--------|---------|---------|---------|----------------|--------------|-----------|---------|--------|
> | EGNO | 9.35 | 58.09 | 4.60 | **12.82** | 0.39 | 0.87 | 10.95 | 0.58 |
> | **SGDD-EGNO** | **7.75** | **13.65** | **4.00** | 13.08 | **0.36** | **0.85** | **8.34** | **0.52** |
> |  | (+17.1%) | (+76.5%) | (+13.0%) | (−2.0%) | (+7.6%) | (+2.2%) | (+23.8%) | (+10.3%) |
> | GFNODE | _7.64_ | _4.89_ | _3.92_ | 12.86 | 0.37 | _0.80_ | _5.00_ | _0.54_ |
> | **SGDD-GFNODE** | **7.38** | **2.23** | **3.64** | **12.72** | **0.34** | **0.79** | **4.97** | **0.52** |
> |  | (+3.4%) | (+54.3%) | (+7.1%) | (+1.0%) | (+8.1%) | (+1.2%) | (+0.6%) | (+3.7%) |
>
> ---
>
>
> | S2T | Aspirin | Benzene | Ethanol | Malonaldehyde | Naphthalene | Salicylic | Toluene | Uracil |
> |--------|---------|---------|---------|----------------|--------------|-----------|---------|--------|
> | EGNO | 7.03 | 30.79 | 3.27 | **10.83** | 0.35 | 0.75 | 4.86 | 0.54 |
> | **SGDD-EGNO** | **6.20** | **8.29** | **2.84** | 11.03 | **0.33** | **0.69** | **3.80** | **0.50** |
> |  | (+11.8%) | (+73.0%) | (+13.1%) | (−1.8%) | (+5.7%) | (+8.0%) | (+21.8%) | (+7.4%) |
> | GFNODE | 6.18 | 2.26 | 2.86 | 11.03 | 0.32 | 0.66 | 2.33 | **0.43** |
> | **SGDD-GFNODE** | **5.66** | **1.15** | **2.66** | **10.91** | **0.27** | **0.60** | **2.32** | **0.43** |
> |  | (+8.4%) | (+49.1%) | (+6.9%) | (+1.0%) | (+15.6%) | (+9.0%) | (+0.4%) | (0.0%) |
>
>
> ### Table A.4. Comparison on the Motion Capture dataset with our own implementations
>
> | S2S | Walk | Run |
> |--------|------|------|
> | EGNO | 11.9 | 37.9 |
> | **SGDD-EGNO** | **6.7** | **28.2** |
> |  | (+43.3%) | (+25.5%) |
>
> | S2T | Walk | Run |
> |--------|------|------|
> | EGNO | 5.5 | 17.6 |
> | **SGDD-EGNO** | **3.2** | **13.5** |
> |  | (+41.5%) | (+23.2%) |
>
>
>
> ### **Table : Comparison of EGNO variants and SGDD-EGNO on MD17 (S2S)**
>
> | **S2S**              | **Aspirin** | **Benzene** | **Ethanol** | **Malonaldehyde** | **Naphthalene** | **Salicylic** | **Toluene** | **Uracil** |
> |-------------------------|-------------|-------------|-------------|-------------------|------------------|---------------|-------------|------------|
> | EGNO (implemented)      | 9.35        | 58.09       | 4.60        | 12.82             | 0.39             | 0.87          | 10.95       | 0.58       |
> | EGNO (SGDD setting)     | 9.33        | 58.10       | 4.60        | 12.82             | 0.38             | 0.89          | 10.95       | 0.53       |
> | **SGDD-EGNO**           | **7.75**    | **13.65**   | **4.00**    | **13.08**         | **0.36**         | **0.85**      | **8.34**    | **0.52**   |

---

> ### Author Response · Authors · 2025-11-21
>
> > **W2. Fairness and tuning budgets**
>
> **A.** We thank the reviewer for raising this important point regarding fairness and tuning budgets. To ensure that the comparison is fair and that improvements do not stem from increased model capacity or extended training, we carefully examined the architectural and training settings of the baseline decoders. As described in the revised Implementation Details and Appendix, we strictly follow the official configurations of EGNO and GFNode for all decoder architectures, without introducing any additional capacity. Only training-related hyperparameters were adjusted to fit the SGDD training procedure. This allows us to isolate the contribution of the proposed representation-learning framework from architectural changes in the baselines. We have added a detailed clarification of these settings in the Appendix C.4.
>
> ---
>
> **Clarification for implementation(Appendix C.4)**
> Our framework is implemented to follow the training and architectural settings of EGNO as closely
> as possible, and we refer readers to the official EGNO implementation for additional details. For the
> decoder component, we strictly adopt the architectural configurations of both EGNO and GFNode
> without introducing any additional capacity; only training-related hyperparameters are adjusted to
> accommodate the SGDD training procedure. This ensures that the observed improvements stem
> from our representation-learning framework rather than increased model complexity in the baseline
> decoders
>
>
>
> > **Q2. Frequency weighting & high-frequency regimes**
>
> **A:** We thank the reviewer for this insightful question. We provide additional frequency-conditioned error profiles in Appendix E. Regarding the request to identify regimes in which high-frequency content becomes dominant, we would like to clarify that the systems and tasks considered in this work—long-horizon trajectory prediction in physical dynamics—are empirically and physically known to exhibit predominantly low-frequency behavior. High-frequency–dominant regimes may indeed arise in other settings, such as very short-term prediction of rapidly oscillatory motions, systems with externally induced high-frequency perturbations, or synthetic datasets specifically designed to emphasize localized fluctuations.

---

> > ### Comment · Reviewer_Podc · 2025-11-27
> >
> > I thank the authors for the detailed rebuttal and the revisions.
> > The rebuttal and revision address my three main concerns to a satisfactory degree. The statistical picture is clearer, the fairness of decoder comparisons is significantly better documented, and the spectral diagnostics are richer.
> > I therefore maintain my original overall assessment. This is a solid, technically sound contribution with a coherent idea of spectral privileged-information distillation for dynamics and convincing gains in the regimes it targets.

---

> > > ### Author Response · Authors · 2025-11-27
> > > **Thank you very much for your thoughtful and encouraging assessment.**
> > >
> > > Thank you very much for your thoughtful and encouraging assessment.
> > > We truly appreciate the time and care you devoted to the detailed review, rebuttal evaluation, and revision check.
> > > We are glad that the clarifications and improvements addressed your concerns and that our contributions were received positively. Your constructive feedback was highly valuable in strengthening the paper, and we sincerely thank you for it.

---

### Official Review · Reviewer_Q8zS · 2025-10-31

**Soundness:** 2
**Presentation:** 2
**Contribution:** 2
**Rating:** 2
**Confidence:** 3

**Summary:**

This paper addresses the challenging problem of long-horizon trajectory prediction for physical dynamics, where systems exhibit a complex interplay of low- and high-frequency components. The authors propose a novel framework, SGDD (Spectral-Guided Dynamics Distillation), which leverages privileged information (future trajectories) within a knowledge distillation setup. The core of SGDD is a spectral-guided enhancement module that operates on a unified spatio-temporal graph representation.

**Strengths:**

- Addresses a Core Challenge: The paper directly tackles the critical challenge of disentangling and prioritizing multi-scale dynamics, which is a primary reason why long-horizon prediction is so difficult. The results on low-frequency error reduction are particularly impressive.
- Strong and Consistent Performance: SGDD consistently outperforms very strong, recent baselines (EGNO, GFNode) across multiple diverse and challenging datasets (molecules, human motion, proteins).
- Generality: The framework is designed to be modular. It can be instantiated with different equivariant GNNs as decoders (as shown with EGNO and GFNode), highlighting its flexibility and potential for broad impact.

**Weaknesses:**

- Complexity of the Framework: The overall system is quite complex, involving multiple encoders, a custom spectral module, and a staged training process. This might pose a challenge for reproducibility and for other researchers seeking to adopt the method.
- Choice of Teacher/Student Architecture: The paper uses sophisticated models (STSGNN, GAT) for the teacher and student encoders. A deeper justification for these specific choices, and an analysis of how the performance depends on the capacity of these encoders, would be beneficial.
- Dependence on a Good Basis: The method's success hinges on the quality of the spatio-temporal graph basis derived from the Laplacian. While this is standard, a discussion on potential limitations for systems where the graph structure is dynamic or poorly defined would be valuable.

**Questions:**

1.  The spectral-guided enhancement relies on a truncated basis of the *K* lowest-frequency modes. The ablation in Figure 6 shows that performance is sensitive to *K*. How should one choose an optimal *K* in practice for a new dataset or system? Is there a more adaptive way to select the basis rather than a fixed low-pass filter?
2.  The dual-level alignment (in both spatio-temporal and spectral domains) is shown to be crucial. Could you provide some intuition as to why aligning in both domains is superior to aligning in just one? Does one domain contribute more to the final performance than the other?
3.  The paper focuses on distilling a representation. Have you considered also distilling the final output distribution (i.e., the predicted trajectory itself), as is common in classic knowledge distillation? Would this provide complementary benefits?

---

> ### Author Response · Authors · 2025-11-21
>
> > **W1. Complexity of the framework**
>
> **A :** We appreciate the reviewer’s concern regarding the overall complexity of the proposed framework. However, the framework is not an arbitrary collection of components; it is designed with a single specific goal in mind—leveraging privileged future-trajectory information during training in order to construct a dynamics representation that enables accurate long-horizon prediction. This objective naturally motivates a knowledge-distillation formulation, where a teacher encoder observes the full trajectory and a student encoder learns to approximate the teacher’s dynamics representation using only the initial condition.
>
> Building on this motivation, SGDD introduces two essential design components to address limitations of conventional distillation:
>
> **(1) Dual Alignment**, which provides complementary forms of guidance:
> - **Feature Align** preserves global spatio-temporal structure by aligning representations in the spatio-temporal domain.
> - **Freq Align** captures multi-scale dynamical behavior by aligning spectral coefficients in the joint frequency domain.
>
> These two alignments reinforce each other and together stabilize the distilled representation.
>
> **(2) Spectral-Guided Enhancement (SGE)**, which adaptively reweights spectral components so that the alignment emphasizes the most informative frequencies, improving long-horizon consistency and robustness.
>
> Thus, each component plays a necessary role in enabling the student to recover a stable and informative multi-scale dynamics representation, and removing any of them leads to clear performance degradation, as demonstrated in Table 4 and Table A.7.
>
> **To further support reproducibility and practical adoption**, we have made the full implementation publicly available at
> `https://anonymous.4open.science/r/SGDD-DCEC`,
> and we provide detailed implementation and training procedures in Appendix C.3 and Appendix C.4.
>
> > **W3. Dependence on a good basis**
>
> **A.** Thank you for raising this important point regarding the reliance on a Laplacian-derived spatio-temporal basis. Our current formulation assumes a fixed graph structure, and thus the basis remains stable throughout the trajectory. For systems with dynamic graph structures where connectivity evolves over time, SGDD is not directly applicable in its current form. However, we agree that extending the framework to time-varying topologies—e.g., by dynamically updating the spatial eigenbasis as adjacency changes—is a promising direction. We have added this discussion as part of the future work in the Conclusion section of the revised manuscript.
>
> Regarding poorly defined or ambiguous graph structures (which we interpret as scenarios with noisy, sparse, or ambiguous connectivity, potentially leading to unstable Laplacians and poor basis quality, as seen in cases of unseen topologies or irrelevant edges in GNN literature), our method assumes clear edge definition criteria (e.g., distance-based thresholds or fixed bonds). By enforcing such criteria during graph construction, the basis remains robust; however, in truly ambiguous cases, preprocessing (e.g., edge pruning via similarity metrics) could mitigate issues.
>
> We sincerely thank the reviewer again for this insightful suggestion, which helped us clarify the scope and limitations of our basis construction.

---

> ### Author Response · Authors · 2025-11-21
>
> > **W2. Choice of teacher/student architecture.**
>
> **A :** We appreciate the reviewer’s request for a deeper justification of the teacher–student architectures.
> We provide our rationale for selecting STSGNN as $E_{\text{dyn}}$ (teacher) and GAT as $E_{\text{init}}$ (student), and we have also added more detailed explanations in Appendix C.3.
>
> **Justification for Choice of STSGNN ($E_{\text{dyn}}$)**
> STSGNN is a sophisticated spatio-temporal graph encoder that performs 2-D joint spectral filtering, enabling spatial and temporal dependencies to be modeled within a unified spectral domain while maintaining stable propagation of both low- and high-frequency components.
> Such frequency-aware modeling aligns well with the characteristics of physical dynamics, where low-frequency structure governs long-term evolution and high-frequency variations capture short-term fluctuations.
> For these reasons, we adopt STSGNN as our dynamics encoder, as it offers the frequency-resolved spatio-temporal representations required for modeling physical systems.
>
> **Justification for Choice of GAT ($E_{\text{init}}$)**
> GAT is the simplest attention-based graph encoder capable of projecting an initial state into a higher-dimensional temporal representation through multi-head attention.
> Moreover, its lightweight architecture and minimal inductive bias help avoid overfitting or underfitting issues that may arise with heavier encoders, making it a suitable choice for the initial encoder within our SGDD framework.
>
> Moreover, **we expanded our analysis by implementing additional encoder variants**---STGCN as an alternative dynamics encoder and GINE or Transformer as alternative initial encoders---and evaluating all combinations under identical training settings.
> The new ablation results (Table 5) show that changes to $E_{\text{init}}$ induce larger performance variation than changes to $E_{\text{dyn}}$.
> GINE yields performance comparable to GAT, whereas Transformer exhibits degradation on several molecules due to its substantially larger parameter count, which leads to overfitting or underfitting under the same optimization configuration.
>
> These findings indicate that SGDD is moderately robust to reasonable encoder substitutions (e.g., GAT ↔ GINE or STSGNN ↔ STGCN), while heavier encoders may require different training strategies to realize their potential.
> Overall, this analysis clarifies how encoder capacity affects performance and further supports the architectural choices made in the main paper. This ablation study evaluating various encoder configurations is presented in Section 4.4.3.
>
> ---
>
> ### **Table 5. MSE (×10⁻²) of SGDD-EGNO across different encoder combinations on MD17 (S2S)**
>
> | **$E_{\text{init}}$** | **$E_{\text{dyn}}$** | **Aspirin** | **Ethanol** | **Naphthalene** | **Salicylic** | **Uracil** |
> |------------------------|------------------------|-------------|-------------|------------------|----------------|-------------|
> | GAT (142K)            | STSGNN (397K)          | 7.75        | 4.00        | 0.36             | 0.85           | 0.52        |
> | GAT (142K)            | STGCN (151K)           | 7.97        | 4.08        | 0.36             | 0.84           | 0.52        |
> | GINE (336K)           | STSGNN (397K)          | 7.85        | 4.21        | 0.36             | 0.86           | 0.53        |
> | GINE (336K)           | STGCN (151K)           | 8.02        | 4.15        | 0.36             | 0.86           | 0.52        |
> | Transformer (541K)    | STSGNN (397K)          | 8.90        | 4.12        | 0.36             | 0.91           | 0.53        |
> | Transformer (541K)    | STGCN (151K)           | 8.69        | 4.27        | 0.36             | 0.94           | 0.53        |

---

> ### Author Response · Authors · 2025-11-21
>
> > **Q1. Sensitivity to the truncation parameter $K$ and how to choose it**
>
> **A.** We appreciate the reviewer’s insightful observation regarding the sensitivity of our method to the truncation parameter $K$.
> To more thoroughly address this point, we have conducted additional experiments to analyze the impact of $K$. Specifically, we performed experiments on two datasets that were not included in the original submission—*MD17 (Aspirin)* (Table A.10)and the *Protein equilibrium trajectory dataset*(Table A.11)—by varying $K$ according to each dataset’s total number of available frequency components.
>
> For datasets with relatively few modes (e.g., *MD17-Aspirin* with 104 modes and *Motion Capture* with 155 modes), the performance exhibits noticeable sensitivity to the choice of $K$.
> In contrast, for the *Protein* dataset (3,420 modes), the results remain stable across a broader range.
> Across all datasets, however, we consistently observe that values near $K = 64$ achieve the strongest performance.
> This aligns with the behavior of physical dynamics, where long-term evolution is largely governed by low-frequency components.
>
> These findings suggest that setting $K \approx 64$ provides a reliable initial choice for new datasets without requiring extensive hyperparameter tuning.
>
> We also note that choosing $K$ adaptively is a promising direction. While we did not explore these mechanisms in the present work, we believe that incorporating a learnable or data-driven selection of $K$ is a natural and valuable extension for future research.
>
> ### **Table A.9.  Motion Capture K Ablation (SGDD–EGNO)**
> | K        | Walk (S2S)      | Run (S2S)       |
> |----------|------------------|------------------|
> | 16       | 9.79 ± 0.18      | 38.82 ± 0.38     |
> | 32       | 8.03 ± 0.05      | 31.88 ± 0.04     |
> | 64       | **6.74 ± 0.00**      | **28.21 ± 0.00**     |
> | full(155)  | 7.26 ± 0.02      | 28.18 ± 0.18     |
>
> | K        | Walk (S2T)      | Run (S2T)       |
> |----------|------------------|------------------|
> | 16       | 4.70 ± 0.10      | 17.99 ± 1.39     |
> | 32       | 3.96 ± 0.06      | 15.37 ± 1.16     |
> | 64       | **3.22 ± 0.05**      | **13.52 ± 0.89**     |
> | 128      | 3.29 ± 0.05      | 14.39 ± 1.05     |
> | full(155)  | 3.46 ± 0.06      | 13.39 ± 0.89     |
>
>
> ### **Table A.10.  MD17(Aspirin) K Ablation (SGDD–EGNO)**
> | K        | Aspirin (S2S) |
> |----------|----------------|
> | 16       | 6.33           |
> | 32       | 6.32           |
> | 64       | **6.20**           |
> | full(104)  | 6.26           |
>
> | K        | Aspirin (S2T) |
> |----------|----------------|
> | 16       | 8.04           |
> | 32       | 8.06           |
> | 64       | **7.84**           |
> | full(104)  | **7.84**           |
>
>
> ### **Table A.11.  Protein K Ablation (SGDD–EGNO)**
> | K        | Protein (S2S) |
> |----------|------------|
> | 64       | **1.74**       |
> | 128      | 1.75       |
> | 256      | 1.75       |
> | 512      | 1.75       |
>
> > **Q3. Could output-level distillation (trajectory-level KD) provide complementary benefits?**
>
> **A.** Thank you for this excellent suggestion on incorporating output-level distillation.
> In classical KD for regression tasks, it is common to distill the teacher's predicted values via MSE loss, providing direct supervision to the student. However, in SGDD, the teacher and student act as encoders extracting dynamics representations (**$z^{sg}_{\text{init}}$**, **$z^{sg}_{\text{dyn}}$**), which serve as inductive biases for a decoder that performs the final trajectory prediction.
>
> This structure makes direct distillation of the final output (predicted trajectory) less suitable, as the encoders do not generate trajectories themselves. Distilling outputs would require modifying the architecture to have task-specific teachers/students. While this might offer complementary benefits, it could deviate from our core emphasis on **frequency-aware representation learning**.
>
> We appreciate the idea and will explore potential integrations in future work.

---

> ### Author Response · Authors · 2025-11-21
>
> > **Q2. Why is dual-level alignment superior, and does one domain matter more?**
>
> **A.** We appreciate the opportunity to clarify the intuition behind our design. As described in Section 3.4, the dual-level alignment operates in two complementary domains: the **spatio-temporal feature domain (Feature Align)**, which aligns the raw dynamics representations to capture integrated spatial interactions and temporal dependencies, and the **spectral domain (Freq Align)**, which aligns the frequency coefficients to emphasize key low- and high-frequency components for long-horizon stability and precision.
>
> Aligning in both domains is superior to one alone because the **spatio-temporal domain preserves holistic structural patterns**, while the **spectral domain disentangles multiscale frequencies** (e.g., global trends vs. local oscillations). Relying on only one domain risks overlooking either **raw spatio-temporal entanglement** or **frequency-specific nuances**, leading to suboptimal distillation and prediction.
>
> Regarding relative contributions, as shown in Table 4, there is **no consistent superiority** between using only spatio-temporal alignment or only spectral alignment. However, using **both simultaneously yields the best performance** overall. This suggests that the two alignments operate in a **mutually complementary manner**, where spatio-temporal alignment provides a **robust inductive bias for overall structure**, and spectral alignment **refines frequency prioritization** to mitigate noise and instability.
>
>
> ### **Table 4. Ablation Study (SGDD–EGNO)**
>
> | Freq Align | Feature Align | SGE | Ethanol | Malonaldehyde | Toluene | Mocap-Walk | Mocap-Run |
> |-----------|---------------|-----|---------|---------------|---------|------------|-----------|
> | ✓ | ✓ | ✓ | **2.84** | **11.03** | **3.80** | **2.95** | 12.98 |
> | ✓ | ✓ | – | 2.90 | 11.04 | 4.18 | 4.04 | **12.61** |
> | ✓ | – | ✓ | 2.89 | 11.11 | 4.86 | 3.30 | 13.01 |
> | – | ✓ | ✓ | 2.85 | 11.06 | 4.65 | 3.26 | 14.37 |

---

### Official Review · Reviewer_nHYS · 2025-10-31

**Soundness:** 2
**Presentation:** 3
**Contribution:** 2
**Rating:** 4
**Confidence:** 3

**Summary:**

This paper proposes a novel framework, Spectral-Guided Dynamics Distillation (SGDD), for long-horizon prediction of 3D particle trajectories. The method is designed to tackle the challenge of modeling complex spatio-temporal dynamics that involve a mix of low- and high-frequency components. The core of SGDD is a knowledge distillation setup where a "teacher" encoder has access to the full future trajectory (privileged information) during training, while a "student" encoder only sees the initial state. Both encoders produce a spatio-temporal representation, which is then refined by a "spectral-guided enhancement" module. This module projects the representation onto a joint spatio-temporal basis (derived from graph Laplacians), applies learnable weights to different frequency components, and reconstructs the representation. The student is trained to mimic the teacher's enhanced representation, effectively distilling the knowledge of future dynamics into a model that operates only on the initial state at inference time. The distilled representation is then fed to a decoder (e.g., EGNO or GFNode) to generate the trajectory. The authors demonstrate state-of-the-art performance on molecular dynamics, protein dynamics, and human motion datasets.

**Strengths:**

1.  **Strong Empirical Performance:** The SGDD framework consistently achieves state-of-the-art results across multiple diverse and challenging benchmarks, demonstrating its effectiveness in practice.
2.  **Effective Use of Privileged Information:** The knowledge distillation setup provides an elegant way to incorporate information from future trajectories during training to improve the representation learned from only the initial state.
3.  **Addresses a Key Challenge:** The paper tackles the important and difficult problem of long-horizon forecasting in systems with complex, multi-scale dynamics.

**Weaknesses:**

1.  **Methodological Ambiguity:** The construction of the spatio-temporal graph and, more importantly, the process of projecting the initial-state representation onto the spatio-temporal basis are not clearly explained. This is a critical flaw that hinders understanding and reproducibility.
2.  **Insufficient Ablation Studies:** The ablation studies do not adequately isolate the contribution of the core novelty—the spectral-guided enhancement. A baseline that uses the same distillation framework but omits the spectral module is essential to validate the paper's central claim.
3.  **Limited Novelty of Components:** The constituent ideas (knowledge distillation with future-as-PK, graph spectral analysis) are not new. The contribution is in their combination, but the significance of this combination is not fully substantiated due to the aforementioned weaknesses.
4.  **Complexity:** The proposed framework is highly complex, involving two separate encoders, a custom spatio-temporal basis construction, spectral projection, re-weighting, reconstruction, and a two-stage training process. This complexity makes it difficult to dissect the reasons for its success and may limit its practical adoption.

**Questions:**

1.  Could you please provide a precise mathematical definition of the temporal graph `G_t` and its Laplacian `L_t`? How are the temporal connections established?
2.  Please provide a detailed explanation of how the representation `z_init`, derived from the single initial graph `G_0`, is prepared for projection onto the spatio-temporal basis `B_K`. What are the exact "projection and expansion" steps mentioned in lines 202-204?
3.  Would it be possible to provide results for an ablation study where knowledge distillation is performed directly on the spatio-temporal representations (`z_dyn` and `z_init`) without the spectral-guided enhancement module? This seems crucial for proving that the spectral component is the key to your method's success.
4.  The truncation parameter `K` is a key hyperparameter. The ablation in Figure 6 shows its impact, but how was `K` chosen for the main experiments on the MD17 and Protein datasets? Is it a fixed value, or was it tuned per dataset? How sensitive are the results to this choice?

---

> ### Author Response · Authors · 2025-11-21
>
> > **W1. Methodological Ambiguity**
>
> **A.** We thank the reviewer for pointing out this ambiguity.
> We have revised the manuscript to more clearly describe both (i) how the spatio-temporal graph is constructed and (ii) how the initial-state representation is projected onto the spatio-temporal basis. Additional explanations have been added to the main text as well as the supplementary material to improve clarity and reproducibility. We appreciate the reviewer for bringing this to our attention, and the updated revisions are provided below.
>
> ---
>
> **(i) Appendix A.2 — Spatio-temporal graph construction**
>
> In our formulation, the spatio-temporal graph \(\mathcal{G}_{st}\) describes both particle interactions and temporal evolution by linking particles within each time step as well as across adjacent time steps. This structure can be viewed as the combination of:
>
> - a **spatial graph** that captures particle connectivity
> - a **temporal graph** that captures sequence order
>
> Each admits its own Laplacian and spectral basis.
>
> #### **Spatial graph**. The spatial graph is defined as $\mathcal{G}_{spatio} = (V, E_s)$. It is fixed across time and encodes the physical connectivity among particles. Its normalized Laplacian is $L_s$ with spatial eigenbasis: $U_s$.
>
> #### **Temporal graph**. The temporal graph is $\mathcal{G}_{temp} = (V_t, E_t)$ with $V_t = \{1, \dots, T\}$. Temporal continuity is modeled using a 1D chain: $E_t = \{(i, i+1) \mid 1 \le i < T\}$. Its normalized Laplacian produces the temporal eigenbasis: $U_t$.
>
> ---
>
> **(ii) Appendix C.3.2 — Implementation Detail of Initial Encoder**
>
>
> We begin with the initial node features:
>
> $$
> x_0 = [Z_0, h] \in \mathbb{R}^{N \times (6+d)}
> $$
>
> For a GAT layer with \(T\) attention heads:
>
> $$
> h_{i,t}^{(0)} = W_t^{(0)} x_{0,i}
> $$
>
> Neighborhood aggregation:
>
> $$
> \tilde{h}\_{i,t}^{(0)} =
> \sum\_{j \in \mathcal{N}(i)}
> \alpha\_{ij,t}^{(0)} \, h\_{j,t}^{(0)}
> $$
>
> Concatenating the heads:
>
> $$
> x\_{1,i} = \Vert_{t=1}^{T} \tilde{h}\_{i,t}^{(0)}
> \in \mathbb{R}^{T d\_z}
> $$
>
> Thus:
>
> $$
> x_1 \in \mathbb{R}^{N \times (T d_z)}
> $$
>
> General layer update:
>
> $$
> x\_{\ell+1,i}
> =\Vert_{t=1}^{T}
> \left(
> \sum\_{j \in \mathcal{N}(i)}
> \alpha\_{ij,t}^{(\ell)}
> W\_t^{(\ell)} x\_{\ell,j}
> \right)
> $$
>
> So:
>
> $$
> x_{\ell+1} \in \mathbb{R}^{N \times (T d_z)}
> $$
>
> Final output:
>
> $$
> x_L \in \mathbb{R}^{N \times (T d_z)}
> $$
>
> Reshaped as:
>
> $$
> \text{reshape}(x_L) =
> z_{\text{init}} \in \mathbb{R}^{N \times T \times d_z}
> $$
>
> ---

---

> ### Author Response · Authors · 2025-11-21
>
> > **W2 & W3 (& Q3). Insufficient ablation studies & Limited novelty of components**
>
> **A.** We thank the reviewer for raising concerns regarding (W2) the need for a clearer ablation isolating SGE and (W3) the perceived limited novelty of the components. Since these points are related, we address them jointly using Table 4.
>
> First, the experiment without Spectral-Guided Enhancement corresponds to removing the learnable spectral weights, which is already included in the second row of Table 4 (now relabeled “SGE”). In this variant, spectral weights are disabled while all other components remain identical. Removing SGE leads to clear performance drops—for example, **+10% on Toluene**, **+37% on Mocap-Walk**, and **+2% on Ethanol**. Table A.7 further shows similar patterns for SGDD-GFNode (e.g., **+24.4% on Aspirin**, **+47.8% on Benzene**, **+32.7% on Mocap-Walk**, **+22.3% on Mocap-Run**), demonstrating that adaptive spectral reweighting is a core component of SGDD.
>
> Second, although our framework builds on knowledge distillation and spectral analysis, it is not a simple combination of the two. To make distillation effective when only the initial state is available, we introduce **Dual Alignment** and **SGE**, designed to interact synergistically. As shown in Table 4, variants using only Feature Align or only Freq Align perform noticeably worse, and full gains appear only when both alignments and SGE operate together. This shows that SGDD’s effectiveness arises from a coordinated design rather than stacking known techniques.
>
> ---
>
> ### **Table 4. Ablation Study (SGDD–EGNO)**
>
> | Freq Align | Feature Align | SGE | Ethanol | Malonaldehyde | Toluene | Mocap-Walk | Mocap-Run |
> |-----------|---------------|-----|---------|---------------|---------|------------|-----------|
> | ✓ | ✓ | ✓ | **2.84** | **11.03** | **3.80** | **2.95** | 12.98 |
> | ✓ | ✓ | – | 2.90 | 11.04 | 4.18 | 4.04 | **12.61** |
> | ✓ | – | ✓ | 2.89 | 11.11 | 4.86 | 3.30 | 13.01 |
> | – | ✓ | ✓ | 2.85 | 11.06 | 4.65 | 3.26 | 14.37 |
>
> ---
>
> ### **Table A.7. Ablation Study on SGE (SGDD–GFNode)**
>
> | Freq Align | Feature Align | SGE | Aspirin | Benzene | Ethanol | Malonaldehyde | Naphthalene | Salicylic | Toluene | Uracil | Mocap-Walk | Mocap-Run |
> |-----------|---------------|-----|---------|---------|---------|---------------|--------------|-----------|---------|--------|------------|-----------|
> | ✓ | ✓ | ✓ | 5.66 | 1.15 | 2.66 | 10.91 | 0.27 | 0.60 | 2.32 | 0.43 | 3.03 | 16.08 |
> | ✓ | ✓ | – | 7.04 | 1.70 | 2.71 | 10.94 | 0.25 | 0.63 | 2.18 | 0.43 | 4.02 | 19.66 |
>
>
> > **W4. Complexity**
>
> **A.** We appreciate the reviewer’s concern regarding the overall complexity of the proposed framework. However, the framework is not an arbitrary collection of components; it is designed with a single specific goal in mind—leveraging privileged future-trajectory information during training in order to construct a dynamics representation that enables accurate long-horizon prediction. This objective naturally motivates a knowledge-distillation formulation, where a teacher encoder observes the full trajectory and a student encoder learns to approximate the teacher’s dynamics representation using only the initial condition.
>
> Building on this motivation, SGDD introduces two essential design components to address limitations of conventional distillation:
>
> **(1) Dual Alignment**, which provides complementary forms of guidance:
> Feature Align preserves global spatio-temporal structure by aligning representations in the spatio-temporal domain.
> Freq Align captures multi-scale dynamical behavior by aligning spectral coefficients in the joint frequency domain.
> These two alignments reinforce each other and together stabilize the distilled representation.
>
> **(2) Spectral-Guided Enhancement (SGE)**, which adaptively reweights spectral components so that the alignment emphasizes the most informative frequencies, improving long-horizon consistency and robustness.
>
> Thus, each component plays a necessary role in enabling the student to recover a stable and informative multi-scale dynamics representation, and removing any of them leads to clear performance degradation, as demonstrated in Table 4 and Table A.7.

---

> ### Author Response · Authors · 2025-11-21
>
> > **Q1. Precise mathematical definition of temporal graph $\mathcal{G}_t$ and Lapacian $L_t$**
>
> **A.** We thank the reviewer for requesting a precise definition.
> The temporal graph $\mathcal{G}_t = (V_t, E_t)$ is defined over the sequence of time indices
> $V_t = \{1, \dots, T\}$.
>
> To model temporal continuity, we adopt the standard 1D chain construction in which each time step is connected to its previous and next steps:
>
> $$
> E_t = \{(i, i+1) \mid 1 \le i < T\}.
> $$
>
> The adjacency matrix $A_t \in \mathbb{R}^{T \times T}$ is defined as:
>
> $$
> (A_t)_{ij} =
> \begin{cases}
> 1, & |i - j| = 1, \\
> 0, & \text{otherwise}.
> \end{cases}
> $$
>
> The diagonal degree matrix \(D_t\) is:
>
> $$
> (D\_t)_{ii} = \sum\_{j=1}^{T} (A\_t)\_{ij}
> $$
>
> The normalized temporal Laplacian is:
>
> $$
> L_t = I - D_t^{-1/2} A_t D_t^{-1/2}.
> $$
>
> The temporal eigenbasis $U_t$ is obtained via the eigendecomposition:
>
> $$
> L_t = U_t \Lambda_t U_t^\top.
> $$
>
> We have added this detailed explanation to Appendix A.2 for clarity.
>
> > **Q2. Preparation of $z_{\text{init}}$ and the projection.**
>
> **A.** Thank you for pointing out the need for more detail on this procedure.
>
> The exact steps proceed as follows: We begin with the initial node features.
>
> $$
> x_0 = [\text{Z}_{0}, \text{h}] \in \mathbb{R}^{N \times (6 + d)}.
> $$
>
> For a GAT layer with \(T\) attention heads, the \(t\)-th head computes
>
> $$
> h_{i,t}^{(0)} = W_{t}^{(0)} x_{0,i},
> $$
>
> and performs neighborhood aggregation as
>
> $$
> \tilde{h}\_{i,t}^{(0)}
> = \sum\_{j \in \mathcal{N}(i)} \alpha\_{ij,t}^{(0)} \, h\_{j,t}^{(0)},
> $$
>
> where $\alpha_{ij,t}^{(0)}$ denotes the attention coefficient for head \(t\).
>
> Concatenating all head outputs yields
>
> $
> x\_{1,i}=
> \Vert\_{t=1}^{T}
> \tilde{h}\_{i,t}^{(0)}
> \in \mathbb{R}^{T d\_z}.
> $
>
> Thus, the full output of the first layer satisfies
>
> $$
> x_1 \in \mathbb{R}^{N \times (T d_z)}.
> $$
>
> For a general \(l\)-th layer $(l \ge 0$), the update rule is
>
> $$
> x\_{\ell+1,i}
> =\Vert\_{t=1}^{T}
> \left(
> \sum\_{j \in \mathcal{N}(i)}
> \alpha\_{ij,t}^{(\ell)} \,
> W\_{t}^{(\ell)} x\_{\ell,j}
> \right)
> $$
>
> and therefore
>
> $$
> x_{\ell+1}
> \in \mathbb{R}^{N \times (T d_z)}.
> $$
>
> Finally, the output of the last layer,
>
> $$
> x_L \in \mathbb{R}^{N \times (T d_z)},
> $$
>
> is reshaped by interpreting the \(T\) heads as temporal channels, yielding
>
> $$
> \mathrm{reshape}(x_L) = z_{\text{init}}
> \in \mathbb{R}^{N \times T \times d_z}.
> $$

---

> ### Author Response · Authors · 2025-11-21
>
> > **Q4. Truncation parameter \(K\) and its sensitivity**
>
> **A:** We sincerely thank the reviewer for the insightful comment regarding the choice and sensitivity of the truncation parameter $K$. Following your suggestion, we have conducted additional experiments to analyze the impact of $K$. Specifically, we performed experiments on two datasets that were not included in the original submission—*MD17 (Aspirin)* and the *Protein equilibrium trajectory dataset*—by varying $K$ according to each dataset’s total number of available frequency components.
>
> For datasets with relatively few modes (e.g., *MD17-Aspirin* with 104 modes and *Motion Capture* with 155 modes), the performance exhibits noticeable sensitivity to the choice of $K$.
> In contrast, for the *Protein* dataset (3,420 modes), the results remain stable across a broader range.
> Across all datasets, however, we consistently observe that values near $K = 64$ achieve the strongest performance.
> This aligns with the behavior of physical dynamics, where long-term evolution is largely governed by low-frequency components.
>
> These findings suggest that setting $K \approx 64$ provides a reliable initial choice for new datasets without requiring extensive hyperparameter tuning.
> We have added the extended results and discussion to the Appendix D.0.2 for completeness.
>
> ---
>
>
> ### **Table A.9.  Motion Capture K Ablation (SGDD–EGNO)**
> | K        | Walk (S2S)      | Run (S2S)       |
> |----------|------------------|------------------|
> | 16       | 9.79 ± 0.18      | 38.82 ± 0.38     |
> | 32       | 8.03 ± 0.05      | 31.88 ± 0.04     |
> | 64       | **6.74 ± 0.00**      | **28.21 ± 0.00**     |
> | full(155)  | 7.26 ± 0.02      | 28.18 ± 0.18     |
>
> | K        | Walk (S2T)      | Run (S2T)       |
> |----------|------------------|------------------|
> | 16       | 4.70 ± 0.10      | 17.99 ± 1.39     |
> | 32       | 3.96 ± 0.06      | 15.37 ± 1.16     |
> | 64       | **3.22 ± 0.05**      | **13.52 ± 0.89**     |
> | 128      | 3.29 ± 0.05      | 14.39 ± 1.05     |
> | full(155)  | 3.46 ± 0.06      | 13.39 ± 0.89     |
>
>
> ### **Table A.10.  MD17(Aspirin) K Ablation (SGDD–EGNO)**
> | K        | Aspirin (S2S) |
> |----------|----------------|
> | 16       | 6.33           |
> | 32       | 6.32           |
> | 64       | **6.20**           |
> | full(104)  | 6.26           |
>
> | K        | Aspirin (S2T) |
> |----------|----------------|
> | 16       | 8.04           |
> | 32       | 8.06           |
> | 64       | **7.84**           |
> | full(104)  | **7.84**           |
>
>
> ### **Table A.11.  Protein K Ablation (SGDD–EGNO)**
> | K        | Protein (S2S) |
> |----------|------------|
> | 64       | **1.74**       |
> | 128      | 1.75       |
> | 256      | 1.75       |
> | 512      | 1.75       |

---

### Author Response · Authors · 2025-11-21
**General response to all Reviewers**

We would like to sincerely thank all reviewers for their thoughtful feedback and constructive suggestions. The comments greatly helped us improve the clarity and overall quality of the manuscript. We are also encouraged that the reviewers highlighted several strengths of our work, including its **well-founded and conceptually clear treatment of multi-scale and low-frequency dynamics** (Reviewers Q8zS, Podc, 6vVc), the **sound and elegant use of privileged information for distillation** (Reviewers nHYS, Podc, 6vVc), the **strong empirical performance and broad applicability** of the framework (Reviewers nHYS, Q8zS), and the **clear writing, well-structured presentation, and modular design** (Reviewers Q8zS, 6vVc).

We are fully prepared to address any concerns or questions raised and welcome the opportunity to offer further clarification where needed. Below, we summarize the major revisions made in response to the reviewers’ suggestions, together with pointers to the corresponding sections in the revised manuscript:

1. **Strengthened Ablation Studies:**
   We enhanced the ablation analysis in section 4.4 by providing clearer explanations and deeper interpretation of each component's contribution, making the interactions within the SGDD framework easier to understand.

2. **Broader Encoder Evaluation:**
   We added new ablation experiments using diverse encoder combinations in Table 5 in section 4.4.3, allowing us to analysis the generality and robustness of the SGDD framework across different architectural choices.

3. **Improved Statistical Rigor:**
   For the main MD17 experiments, we conducted multiple independent runs and now report mean and variance to provide stronger statistical reliability. The corresponding updates are shown in Table 1.

For more detailed explanations, please refer to our individual responses to each reviewer. If any additional concerns or questions arise, we would be glad to provide further clarification throughout the discussion phase. We appreciate the opportunity to continue the discussion.

---

### Meta-Review · Area_Chair_ivkq · 2025-12-08

**Summary:**

This paper proposes a method for long-horizon trajectory prediction in physical dynamics, using "privileged" information via future trajectories, and a joint spatio-temporal spectral basis with an adaptive weighting approach. The initial scores were mixed, one reviewer previously raised, one confirmed the 6. The authors have provided a detailed rebuttal.

**Reviewer Concerns:**

The reviewers saw quite a few positive points: empirical results and consistent improvements across datasets; Clear contribution in spectral distillation with privileged information; Effective use of joint spatio-temporal basis and spectral-guided enhancement.

The rebuttal expanded the ablations, and  improved clarity of the submission. The main negative points from reviewer Q8zS seem to primarily be caused by architectural complexity and conceptual about the main concept.

**Reviewer Scores:**

The scores ended up being:
- 6vVc 4 raised to 6
- Podc 6 confirmed
- Q8zS 2 kept
- nHYS 4 kept

For reviewer nHYS, I see a good chance for a raise during a regular rebuttal period.

---

### Decision · Program_Chairs · 2026-01-26

Accept (Poster)